**JCB** Journal of Cell Biology

**TOOLS**

# Genome-wide conditional degron libraries for functional genomics

Eduardo Gameiro[1]*, Karla A. Juárez-Núñez[1]*, Jia Jun Fung[1], Susmitha Shankar[1], Brian Luke[1,2]**, and Anton Khmelinskii[1,3]**

Functional genomics with libraries of knockout alleles is limited to non-essential genes and convoluted by the potential accumulation of suppressor mutations in knockout backgrounds, which can lead to erroneous functional annotations. To address these limitations, we constructed genome-wide libraries of conditional alleles based on the auxin-inducible degron (AID) system for inducible degradation of AID-tagged proteins in the budding yeast *Saccharomyces cerevisiae*. First, we determined that N-terminal tagging is at least twice as likely to inadvertently impair protein function across the proteome. We thus constructed two libraries with over 5,600 essential and non-essential proteins fused at the C-terminus with an AID tag and an optional fluorescent protein. Approximately 90% of AID-tagged proteins were degraded in the presence of the auxin analog 5-Ph-IAA, with initial protein abundance and tag accessibility as limiting factors. Genome-wide screens for DNA damage response factors revealed a role for the glucose signaling factor *GSF2* in resistance to hydroxyurea, highlighting how the AID libraries extend the yeast genetics toolbox.

## Introduction

Characterization of loss-of-function alleles is the most common approach to study gene function. With its simple genetics and a broad range of resources, the budding yeast *Saccharomyces cerevisiae* is an excellent model for functional genomics (Botstein and Fink, 2011). The budding yeast knockout library was the first genome-wide collection of gene deletion strains (Giaever et al., 2002; Winzeler et al., 1999). This resource has enabled hundreds of genome-wide screens for functional profiling of the yeast genome, mapping of genetic interactions, and identification of drug targets and mechanisms of drug action (Giaever and Nislow, 2014).

Despite the wide-ranging impact of the yeast knockout library, this resource has some limitations. First, because ~20% of yeast genes are essential (Giaever et al., 2002), i.e., their deletion is lethal, the knockout library should be complemented with conditional alleles of essential genes for complete coverage. Libraries of essential alleles based on transcriptional repression, decreased expression by reducing gene dosage through the use of heterozygous diploids or by mRNA destabilization, temperature sensitivity, or targeted protein degradation have been applied with different trade-offs, including variability in the rate and extent of downregulation or side effects of the conditions necessary for downregulation (Li et al., 2011; Ben-Aroya et al.,

2008; Breslow et al., 2008; Snyder et al., 2019; Mnaimneh et al., 2004; Kanemaki et al., 2003; Giaever et al., 2002). Another limitation of the knockout library is the masking of gene-specific phenotypes by spontaneous suppressor mutations that can arise in gene deletion strains (Hughes et al., 2000; Teng et al., 2013; van Leeuwen et al., 2016).

Here, we sought to address these limitations by developing genome-wide libraries of conditional alleles based on the AID2 auxin-inducible degron (AID) system for targeted protein degradation (Yesbolatova et al., 2020; Nishimura et al., 2009). The improved AID2 system allows for tightly controlled and rapid degradation of AID-tagged proteins (Nishimura et al., 2009). We show that near-complete degradation of most yeast proteins can be achieved with genome-wide AID libraries and demonstrate an application of these resources in genome-wide screens for factors involved in the DNA damage response.

## Results

### Systematic comparison of N- and C-terminal tagging

We set out to construct genome-wide AID libraries, where each strain expresses a different protein fused to the AID tag. To decide at which terminus to fuse the AID tag so that the

[1]Institute of Molecular Biology, Mainz, Germany; [2]Johannes Gutenberg University Mainz, Institute for Developmental Neurology, Mainz, Germany; [3]Institute for Quantitative and Computational Biosciences, Johannes Gutenberg University Mainz, Mainz, Germany.

*E. Gameiro and K.A. Juárez-Núñez contributed equally to this paper; Correspondence to Anton Khmelinskii: a.khmelinskii@imb-mainz.de

**B. Luke and A. Khmelinskii jointly supervised this work.



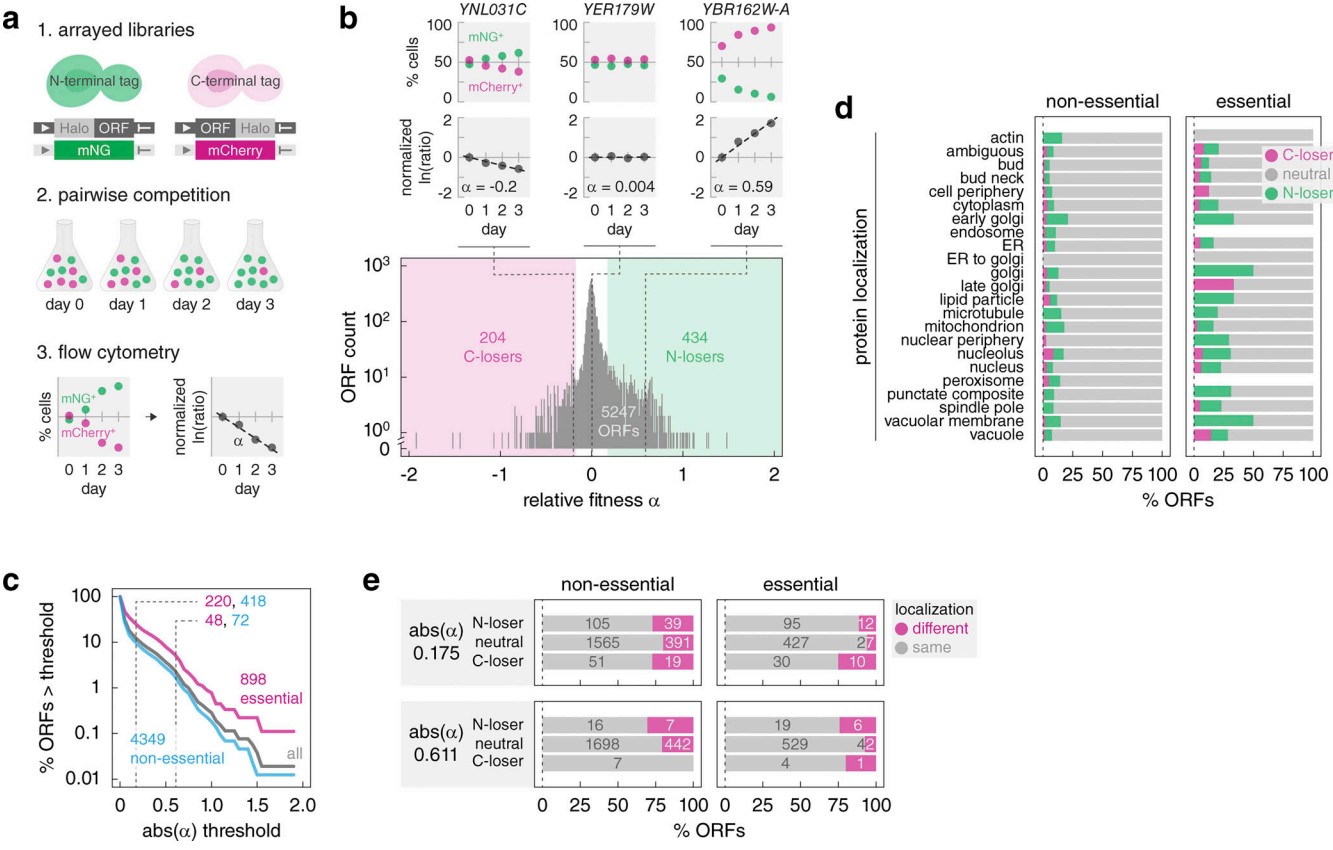

**Figure 1. Fitness impact of N- versus C-terminal protein tagging. (a)** Competition assay to compare the fitness of strains with N- or C-terminally tagged proteins. Pairs of strains with the same ORF tagged at the endogenous chromosomal locus are co-cultured. The percentage of each cell type in the population is followed with flow cytometry using strain-specific fluorescent markers expressed from the constitutive *GPD* promoter (mNG, green, or mCherry, magenta). The slope α of a linear fit to ln(mCherry+ cells/mNG+ cells) over time represents the relative fitness of the two strains (α > 0, competitive advantage of the ORF-Halo strain). **(b)** Genome-wide assessment of N- versus C-terminal protein tagging impact on fitness. Distribution of relative fitness for 5,247 ORFs assayed according to a. ORFs with α < −0.175 are defined as C-losers (C-terminal tagging impairs fitness, magenta area), α > 0.175 – N-losers (green area). Top, percentage of mCherry+ and mNG+ cells and linear fit to the ln(mCherry+/mNG+ cells) over time for three representative ORFs. **(c)** Percentage of ORFs differentially affected by N- versus C-terminal tagging as a function of a relative fitness threshold. Number of ORFs, essential (magenta) or non-essential (blue), differentially affected at two thresholds (dashed lines, abs(α) = 0.175 [5% fitness difference] and 0.611 [20%], Materials and methods) is indicated. **(d)** Frequency of relative fitness differences according to protein localization. Subcellular localization based on fluorescence microscopy imaging of 3,827 strains expressing C-terminally GFP-tagged proteins (Huh et al., 2003). **(e)** Proteins with differential localization of N- and C-terminally GFP tagged variants (Huh et al., 2003; Weill et al., 2018), stratified by gene essentiality and differential fitness according to b. N-loser, neutral, and C-loser ORFs defined at two relative fitness thresholds. Number of ORFs in each group is indicated.

likelihood of tagging artifacts is minimized, we asked whether protein N- or C-termini are generally more amenable to tagging. Although studies of individual proteins have shown that tagging can impair protein localization or function depending on the tagged terminus, a comprehensive comparison of N- and C-terminal tagging is lacking (Weill et al., 2019).

To systematically assess the relative impact of N- and C-terminal tagging, we established a high-throughput competition assay (Fig. 1 a). In this assay, pairs of strains expressing the same protein fused either at the N- or at the C-terminus to a non-fluorescent tag are co-cultured. The frequency of each cell type in the co-cultures is monitored over time by flow cytometry of strain-specific fluorescent markers, the mNeonGreen (mNG) (Shaner et al., 2013) and mCherry (Shaner et al., 2004) fluorescent proteins expressed from a strong constitutive promoter. The relative fitness (α) of the two co-cultured strains is then determined from the change in the relative frequencies of the

two cell types over time (Fig. 1 a, Materials and methods). Under the assumption that tagging is either neutral or impairs fitness, but is very unlikely to improve fitness, α < 0 corresponds to reduced relative fitness of the C-terminally tagged strain and α > 0 indicates reduced relative fitness of the N-terminally tagged one.

We used the high-throughput SWAp-Tag (SWAT) approach (Yofe et al., 2016; Weill et al., 2018; Meurer et al., 2018) to tag open reading frames (ORFs) at endogenous chromosomal loci with a sequence encoding the Halo tag (Los et al., 2008). The SWAT approach makes use of strain arrays with individual ORFs marked with an acceptor module. This module is inserted immediately before the stop codon in C-SWAT strains or after the start codon in N-SWAT strains, except for ORFs encoding proteins with predicted N-terminal signal peptides or N-terminal mitochondrial targeting signals (MTS), where the acceptor module is placed after the predicted signal sequences. Using

semiautomated crossing, the acceptor module can be replaced with practically any tag or regulatory element provided on a donor plasmid (Yofe et al., 2016). We chose the Halo tag due to its size (33 kDa), similar to many commonly used fluorescent protein tags and to the mNG-AID*-3myc tag in the AID-v1 library described below, and lack of evidence for a dominant negative effect on the tagged proteins. Halo tagging using N-SWAT and C-SWAT libraries was efficient, yielding homogeneous strains with over 99% of the cells in most strains expressing the expected fusion protein under the control of its native regulatory elements (Fig. S1 a, Materials and methods). As the resulting Halo-ORF and ORF-Halo libraries are haploid, the ORF tagged in each strain is the only source of that specific protein.

Next, we competed pairs of matched N- and C-terminally tagged strains and determined the relative fitness impact of the Halo tag location (Fig. 1 a). The competition assay yielded reproducible estimates of relative fitness α (Fig. S1 b). The distribution of relative fitness across all assayed pairs was centered at 0.02 ± 0.17 (mean ± SD, $n$ = 5,247 ORFs) (Fig. 1 b and Table S1), suggesting that the impact of the Halo tag was generally independent of the tagged terminus. Importantly, the strain-specific fluorescent markers affected fitness to a similar degree, as indicated by the relative fitness of 0.05 ± 0.01 (mean ± SD, $n$ = 92 technical replicates) for a pair of untagged strains expressing mNG or mCherry (Fig. S1 c). Therefore, the contribution of the fluorescent makers to the detected fitness differences is likely negligible. Based on these observations, we estimated that fitness differences as low as 5% (corresponding to an absolute α = 0.175, Materials and methods) can be measured in the competition assay. Hereafter, ORFs with α > 0.175 are referred to as N-losers, with α < −0.175 as C-losers, and with absolute α ≤ 0.175 as neutral.

Tagging of non-essential ORFs, especially those whose deletion has little to no impact on growth, is unlikely to impair fitness. Accordingly, essential ORFs were more likely to exhibit differential fitness defects in the competition assay (24% compared with 10% of non-essential ORFs) (Fig. 1 c). This suggests that up to a quarter of the proteome is differentially affected by terminal tagging under the assumption that tagging similarly affects essential and non-essential proteins. In addition, tagging at both termini can be detrimental to the function of some proteins. Such cases are likely rare, as no single instance could be detected in triple competition assays for a random set of 45 ORFs, where both tagged variants competed against the wild type (Fig. S1, d and e).

Whereas 155 out of 898 essential ORFs were N-losers, exhibiting reduced relative fitness with N-terminal compared to C-terminal tagging, only 65 essential ORFs were C-losers. This tendency held across the whole dataset, with a total of 434 N-losers and 204 C-losers (Fig. 1 b), indicating that N-terminal tagging is more likely to impair protein function compared with C-terminal tagging. Interestingly, the frequency of differential fitness defects varied between subcellular compartments (Fig. 1 d). Proteins of the golgi, vacuolar membrane, and lipid particle were most likely to be differentially affected by terminal tagging, suggesting that terminal tagging could affect protein localization. Supporting this notion, differential fitness defects

correlated with differential localization of N- and C-terminally tagged proteins (Fig. 1 e). Using datasets of subcellular localizations for most yeast proteins determined with fluorescence microscopy of strains expressing N- and C-terminal GFP fusions (Huh et al., 2003; Breker et al., 2013; Weill et al., 2018), we observed that for essential genes, only 6% of neutral ORFs exhibited different localizations of N- and C-terminally tagged variants, whereas this fraction increased to 11% and 25% for N-losers and C-losers, respectively (Fig. 1 e). The magnitude of this trend is likely underestimated considering that for 636 proteins the N-terminal Halo tag was not placed at the very N-terminus but after the predicted N-terminal signal peptide or mitochondrial targeting signal, to avoid blocking these localization signals and increase the likelihood of correct protein localization (Yofe et al., 2016). Optimized tagging of proteins with these N-terminal localization signals likely also contributes to the lack of correlation between differential fitness defects and the occurrence of terminal localization signals (Fig. S1 f and Table S2). Taken together, we concluded that while N-terminal tagging can be beneficial in individual cases, C-terminal tagging is overall less likely to affect protein localization and function. Nevertheless, further work is needed to understand how the type of tag, its size and biophysical properties, and the linker between the tag and the protein of interest affect protein localization and function across the proteome.

## AID libraries

Based on the above analysis, we constructed genome-wide C-terminally tagged AID libraries. We combined the short AID* tag, which consists of amino acids 71–114 from AtIAA17 (Morawska and Ulrich, 2013; Nishimura et al., 2009), with the AID2 system (Yesbolatova et al., 2020) (Fig. 2 a). The AID2 system relies on the OsTir1 F-box protein with the F74G mutation and a ligand, the auxin analog 5-phenyl-indole-3-acetic acid (5-Ph-IAA), to achieve quick and efficient degradation of AID-tagged proteins at significantly lower ligand concentrations than the original AID system and without basal degradation in the absence of 5-Ph-IAA (Nishimura et al., 2009; Yesbolatova et al., 2020).

Expression of OsTir1(F74G) from the strong constitutive *GPD* promoter resulted in an obvious fitness defect, whereas expression from the weaker *CYC1* and *ADH1* promoters did not affect fitness and conditional expression from the strong *GAL1* promoter had a minor impact on fitness (Fig. S2 a). We decided to use the *GAL1-OsTIR1(F74G)* construct for the AID libraries as the conditional nature of the *GAL1* promoter is likely to limit adaptation to expression of OsTir1(F74G) and the high expression levels of OsTir1(F74G) are less likely to limit degradation of AID-tagged proteins. Using C-SWAT, we tagged each ORF with mNG-AID*-3myc (AID-v1 library, 5,604 tagged ORFs) or with AID*-3myc (AID-v2 library, 5,620 tagged ORFs) and concomitantly introduced the *GAL1pr-OsTIR1(F74G)* expression construct (Fig. 2 b and Fig. S2 b). The 3myc epitope tag can be used to determine protein levels with immunoblotting, whereas the mNG tag in the AID-v1 library allows assessing protein levels with fluorescence measurements. Whereas all strains in the AID-v2 library carry the *OsTIR1* construct, the AID-v1 library

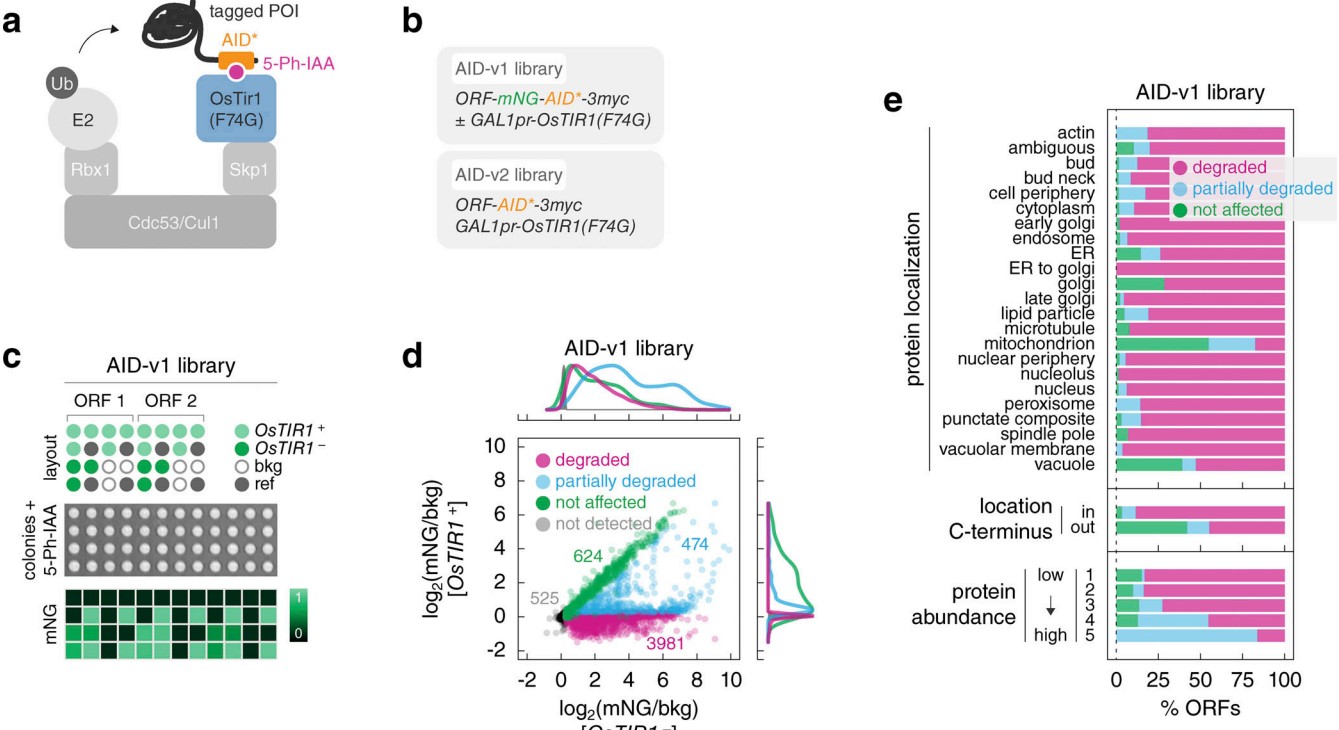

Figure 2. **Efficient protein degradation with genome-wide AID libraries. (a)** Features of the auxin-inducible degron system in the AID libraries. A protein of interest (POI) with a C-terminal AID* tag can be ubiquitinated and degraded in the presence of 5-Ph-IAA upon expression of the F-box protein OsTir1. **(b)** Different tags used in the AID libraries. Expression of *OsTIR1* is controlled by the galactose-inducible *GAL1* promoter. **(c)** Characterization of the AID-v1 library with fluorescence measurements of colony arrays. Top, for each ORF, mNG-AID*-3myc-tagged strains with (+) or without (−) the *GAL1pr-OsTIR1* construct were placed next to each other. A reference strain (ref, with an abundant mNG-tagged protein, for correction of spatial and plate effects) and a strain without a fluorescent tag (bkg, for background correction) were included in each 4 × 4 group. Middle, colony arrays were grown on galactose medium with 1 µM 5-Ph-IAA. Bottom, example heatmap of mNG fluorescence intensities. **(d)** Protein levels (mNG signal in units of background fluorescence, mNG/bkg) in the AID-v1 library determined according to c. Not detected, strains without a detectable mNG signal in the absence of *OsTIR1* (mNG/bkg($OsTIR1^-$) ≤ 1.2). Detectable proteins were classified by the extent of *OsTIR1*-dependent degradation (Materials and methods). Top and right, normalized frequency distributions of protein levels in each category. **(e)** Frequency of *OsTIR1*-dependent degradation phenotypes in the AID-v1 library according to protein localization, location of the C-terminus (cytosol – in, lumen of organelles or extracellular – out) for membrane proteins according to (Kim et al., 2006) or abundance of the mNG-AID*-3myc-tagged proteins.

comprises two strains for each ORF, with or without *OsTIR1* (*OsTIR1⁺* and *OsTIR1⁻*, respectively) (Fig. 2 b).

We assessed the frequency and the extent of AID-dependent degradation across the proteome using mNG fluorescence in the AID-v1 library. Strains were grown as ordered colony arrays on agar plates with galactose and 1 µM 5-Ph-IAA, and their fluorescence was measured with a plate reader to determine the levels of mNG-AID*-3myc-tagged proteins (Fig. 2 c, Materials and methods). We estimated the detection limit of the colony fluorescence assay at ~200 molecules per cell (Fig. S2 c). Protein expression levels in *OsTIR1⁻* strains correlated well with independent estimates of protein abundance (Meurer et al., 2018) (Fig. S2 d), indicating that the AID* tag has little impact on protein levels in the absence of the cognate F-box protein. Out of 5,079 proteins detected in *OsTIR1⁻* strains, 4,455 (~88%) were significantly depleted in *OsTIR1⁺* strains (Fig. 2 d and Table S3). 3,981 proteins could not be detected specifically in the *OsTIR1⁺* background. Hereafter, we will refer to these proteins as degraded, although it is likely that at least in some cases degradation is not complete but the remainder is below the detection limit of our plate reader assay. Nevertheless, 474 proteins were

unequivocally degraded only partially, as they were detectable in the *OsTIR1⁺* background but at reduced levels compared with the *OsTIR1⁻* background (Fig. 2 d). We verified this classification by immunoblotting in time-course experiments. Proteins from the degraded group were depleted to below the detection limit within 60 min of 5-Ph-IAA addition, partially degraded proteins were depleted less or exhibited a degradation-resistant pool, and levels of not affected proteins remained stable over time (Fig. S2 e). The fraction of conditionally degraded proteins (degraded or partially degraded) varied with subcellular localization and was lowest for proteins localized to membrane-bound organelles such as mitochondria, vacuole, golgi, and endoplasmic reticulum (ER) (Fig. 2 e), likely due to inaccessibility of the AID* tag to the cytosolic OsTir1. Supporting this notion, membrane proteins with cytosolic C-termini were more likely to be conditionally degraded compared to those with lumenal or extracellular C-termini (Kim et al., 2006) (Fig. 2 e). Finally, the extent of AID-dependent degradation varied with initial protein abundance, in that highly expressed proteins were more likely to be only partially degraded compared to lowly expressed ones (Fig. 2 e and Fig. S2 f), suggesting saturation of the AID machinery. It

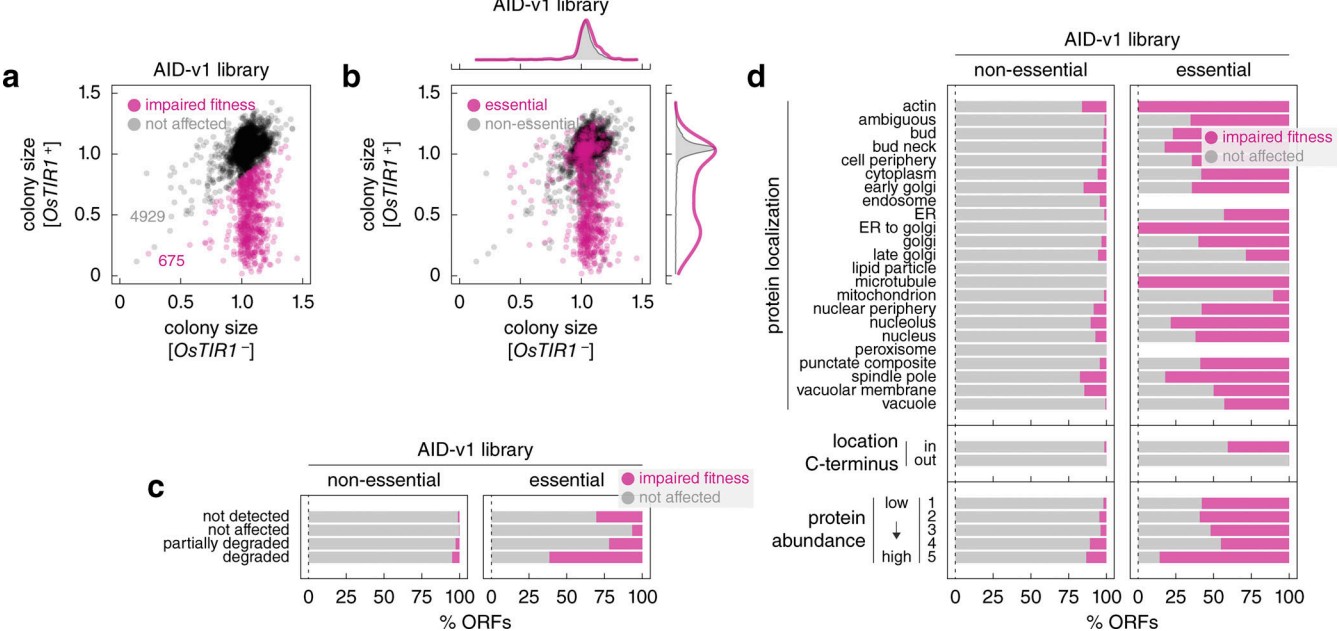

Figure 3. **Fitness defects in the AID libraries upon induction of protein degradation. (a and b)** Normalized colony sizes in the AID-v1 library grown according to Fig. 2 c. **(b)** Essential and non-essential ORFs are marked; top and right, normalized frequency distributions of colony sizes in each category. **(c and d)** Frequency of OsTIR1-dependent fitness defects in the AID-v1 library according to the extent of OsTIR1-dependent degradation phenotypes (c) or according to localization, location of the C-terminus (cytosol – in, lumen of organelles or extracellular – out) or abundance of the mNG-AID*-3myc-tagged proteins (d).

remains to be determined whether OsTir1 or the yeast subunits of the Skp1-cullin-F-box protein (SCF) ubiquitin ligase are limiting factors for the degradation of abundant proteins.

We asked whether AID-dependent protein degradation results in cellular phenotypes. For that, we measured colony sizes in the arrayed AID-v1 library as a readout of strain fitness (Fig. 2 c, Materials and methods). We first quantified the fitness impact of OsTir1 expression by comparing colony sizes of OsTIR1[−] and OsTIR1[+] strains for non-essential ORFs. As degradation of non-essential proteins is not expected to affect fitness, the difference in colony size between OsTIR1[−] and OsTIR1[+] backgrounds can be attributed to OsTir1 expression. On average, the presence of the OsTIR1 construct reduced colony size by 7% (Fig. S3 a). The same difference was observed for the 624 ORFs in the not-affected group (Fig. 2 d and Fig. S3 a), where AID-tagged proteins were not degraded in the OsTIR1[+] background. To determine how AID-dependent protein degradation affects fitness, we then corrected colony sizes of OsTIR1[+] strains by the fitness impact of the OsTIR1 construct (Materials and methods). After the correction, we found that ∼12% of OsTIR1[+] strains (675 out of 5,604) exhibited impaired fitness compared to OsTIR1[−] strains (Fig. 3 a and Table S3). As expected, the frequency of fitness defects was higher for essential proteins. Whereas only ∼4% of OsTIR1[+] strains for non-essential ORFs (176 out of 4,698) had impaired fitness, fitness defects were observed for over 55% of essential ORFs (499 out of 906) (Fig. 3 b). A similar frequency was previously observed with a set of AID alleles constructed for 758 essential ORFs using the original AID system (Snyder et al., 2019). However, over a third of these alleles exhibited fitness defects even in the absence of auxin, which were further compounded by the off-target effects of auxin.

Importantly, the frequency of fitness defects in the AID-v1 library correlated with the extent of protein degradation. Whereas degradation resulted in a fitness defect for over 61% of essential proteins, this fraction dropped to 22% and 7% of essential proteins that were partially degraded or not degraded in an AID-dependent manner, respectively (Fig. 3 c). Interestingly, the degradation of 38% of essential proteins did not result in a fitness defect. We considered three explanations for this observation. First, it is possible that in some cases degradation is incomplete, resulting in low protein levels that are below the detection limit of our assay but are sufficient for viability. As proteins normally expressed at low levels are more likely to fall below the detection limit even when partially depleted, we asked whether strains with essential and degraded proteins but without a fitness defect are enriched in low-abundance proteins. However, there was no difference in protein levels between OsTIR1[−] strains for essential and degraded proteins with and without a fitness defect in the OsTIR1[+] background (Fig. S3 b). Second, we considered the nature of the essential genes in these two groups. Namely, we compared the frequency of core essential genes, which are always required for viability, and conditional essential genes, which vary in essentiality depending on the genetic background or environment (Bosch-Guiteras and van Leeuwen, 2022). Interestingly, the set of essential and degraded proteins without an accompanying fitness defect was enriched in conditional essential genes defined by two independent measures: essentiality across S. cerevisiae natural isolates (Peter et al., 2018) or with bypass suppression interactions in a laboratory strain (van Leeuwen et al., 2020) (Fig. S3 c, odds ratio = 1.6, P value = 0.04 in a Fisher's exact test and odds ratio = 1.7, P value = 0.02, respectively). This suggests that conditional essentiality could explain the observed

lack of fitness defects upon degradation of some essential proteins. Third, it is likely that the fraction of strains with the degradation phenotype is overestimated in our analysis as colony fluorescence usually scales with colony size, which would result in underestimated fluorescence levels for strains with fitness defects (Fung et al., 2022). Indeed, *OsTIR1*+ strains with impaired fitness typically exhibited fluorescence levels below background autofluorescence (Fig. S3, d and e). Finally, the frequency of fitness defects depended on the accessibility of the tag and the initial abundance of the tagged proteins (Fig. 3 d), in agreement with the analysis of the degradation extent (Fig. 2 e). Thus, while the likelihood and the extent of conditional protein degradation in the AID libraries depend on the abundance of the target protein and the accessibility of the AID tag, most yeast proteins can be efficiently degraded with the AID system, resulting in cellular phenotypes consistent with loss of the target protein.

**Screens for DNA damage response factors**
We sought to apply the AID libraries to screen for factors involved in the DNA damage response using sensitivity to genotoxic agents (GAs) as a phenotypic readout. We decided to use the DNA damaging agent methyl methanesulfonate (MMS), the topoisomerase I inhibitor camptothecin (CPT), and the ribonucleotide reductase inhibitor hydroxyurea (HU) as GAs and the AID-v2 library, reasoning that an overall smaller tag (without the mNG moiety) is less likely to inadvertently affect protein function. The library was grown as ordered colony arrays on galactose media with 5-Ph-IAA, a GA or both compounds (Fig. 4 a). First, we observed that treatment of the AID-v2 library with 1 µM 5-Ph-IAA resulted in fitness defects consistent with the AID-v1 library (spearman correlation coefficient ρ = 0.6, Fig. S4 a). These fitness defects similarly correlated with protein essentiality, expression levels, tag accessibility, and extent of AID-dependent degradation (Fig. S4, b–e), indicating that the AID-v1 and AID-v2 libraries perform comparably.

Next, we calculated chemical-genetic interaction (CGI) scores between each AID allele and a GA (Fig. S5 a). We focused on negative CGIs whereby the double perturbation (5-Ph-IAA and GA) results in a stronger fitness defect than expected from the combined effects of the single perturbations (Materials and methods) (Parsons et al., 2004; Baryshnikova et al., 2010b). This led to the identification of 93, 32, and 93 potential resistance factors (CGI score ≤ –0.2 and P value <0.05 in a two-sided *t* test adjusted for multiple testing using the Benjamini-Hochberg method) for MMS, CPT, and HU, respectively (Fig. 4 b and Table S4). There was substantial overlap between the non-essential factors identified here and previous screens with the yeast knockout library (Fig. S5 b) (Parsons et al., 2004; Chang et al., 2002). Nevertheless, AID alleles of most resistance factors identified with knockout strains were not sensitive to the corresponding GAs. We asked whether these factors might not be degraded in the respective AID strains. However, the frequency of ORFs with AID-dependent degradation was similar for screen-specific and shared resistance factors (Fig. S5 b). Differences in growth conditions and definitions of significant fitness defects likely account for the screen-specific CGIs. There were 58

essential genes among the combined 165 potential resistance factors identified in the three screens (Fig. S5 c). CGIs for both essential and non-essential genes were reproducible in spot tests (Fig. S5 d). Additional essential factors could in principle be identified in further screens with reduced 5-Ph-IAA concentrations, which allow determining CGIs for strains that otherwise exhibit no growth in 1 µM 5-Ph-IAA (Fig. S5 e). Taken together, these results validate the screening approach with AID libraries.

Although the three GAs act through distinct mechanisms, all can result in the accumulation of DNA double-strand breaks (DSBs) (Wyatt and Pittman, 2006; Singh and Xu, 2016; Li et al., 2017). Accordingly, shared resistance factors for MMS, CPT, and HU included most members of the *RAD52* epistasis group (Fig. 4, b and c), which are involved in the repair of DSBs (Symington, 2002; Game and Mortimer, 1974; Resnick, 1969). Moreover, most of the identified resistance factors were associated with gene ontology terms related to DNA damage response, transcription, chromatin organization, and cell cycle (Fig. 4 d and Fig. S5 c, Materials and methods). Notably, whereas CPT- and MMS-specific factors were enriched in genes associated with cell cycle progression, DNA damage response, and transcription, HU-specific factors were enriched in components of the endomembrane system, including proteins of the ER and components of endosomal sorting complexes required for transport (Fig. 4 d; Fig. S5 c; and Table S5). This suggests that HU impairs proliferation through multiple mechanisms, consistent with prior reports (Takano and Nakatsukasa, 2023, *Preprint*; Kuong and Kuzminov, 2009; Huang et al., 2016; Musiałek and Rybaczek, 2021).

Finally, we examined *GSF2*, one of the HU-specific resistance factors identified in the screen (Fig. 4 d and Fig. S5 c). Gsf2 is an integral membrane protein localized to the ER that is involved in the biogenesis and export of the hexose transporter Hxt1 from the ER (Sherwood and Carlson, 1999). We confirmed that the *GSF2-AID** strain from the AID-v2 library is sensitive to HU specifically in the presence of 5-Ph-IAA (Fig. 4 e). A *gsf2*Δ mutant from the haploid yeast knockout library (Giaever et al., 2002; Winzeler et al., 1999) was not sensitive to HU. In contrast, a *gsf2*Δ strain generated by sporulation of a heterozygous diploid knockout exhibited HU sensitivity similar to the *GSF2-AID** strain (Fig. 4 e). This suggests that additional mutations acquired by the knockout library strain are masking the HU sensitivity phenotype. To gain insights into the role of *GSF2* in resistance to HU, we analyzed how HU affects the cell cycle progression of the *gsf2*Δ mutant. Knockout of *GSF2* did not prevent yeast cells from arresting in the early S-phase in response to HU (Fig. 4 f), suggesting that *GSF2* plays no role in the activation of the DNA replication checkpoint. However, the *gsf2*Δ mutant exhibited delayed cell cycle progression after release from HU (Fig. 4 g). This phenotype was distinct from a strain lacking the checkpoint kinase Rad53, which is essential for recovery from DNA replicational stress (Bell and Labib, 2016; Tercero and Diffley, 2001; Desany et al., 1998). These observations point toward a role of *GSF2* in recovery from the checkpoint arrest but the underlying mechanism remains to be determined.

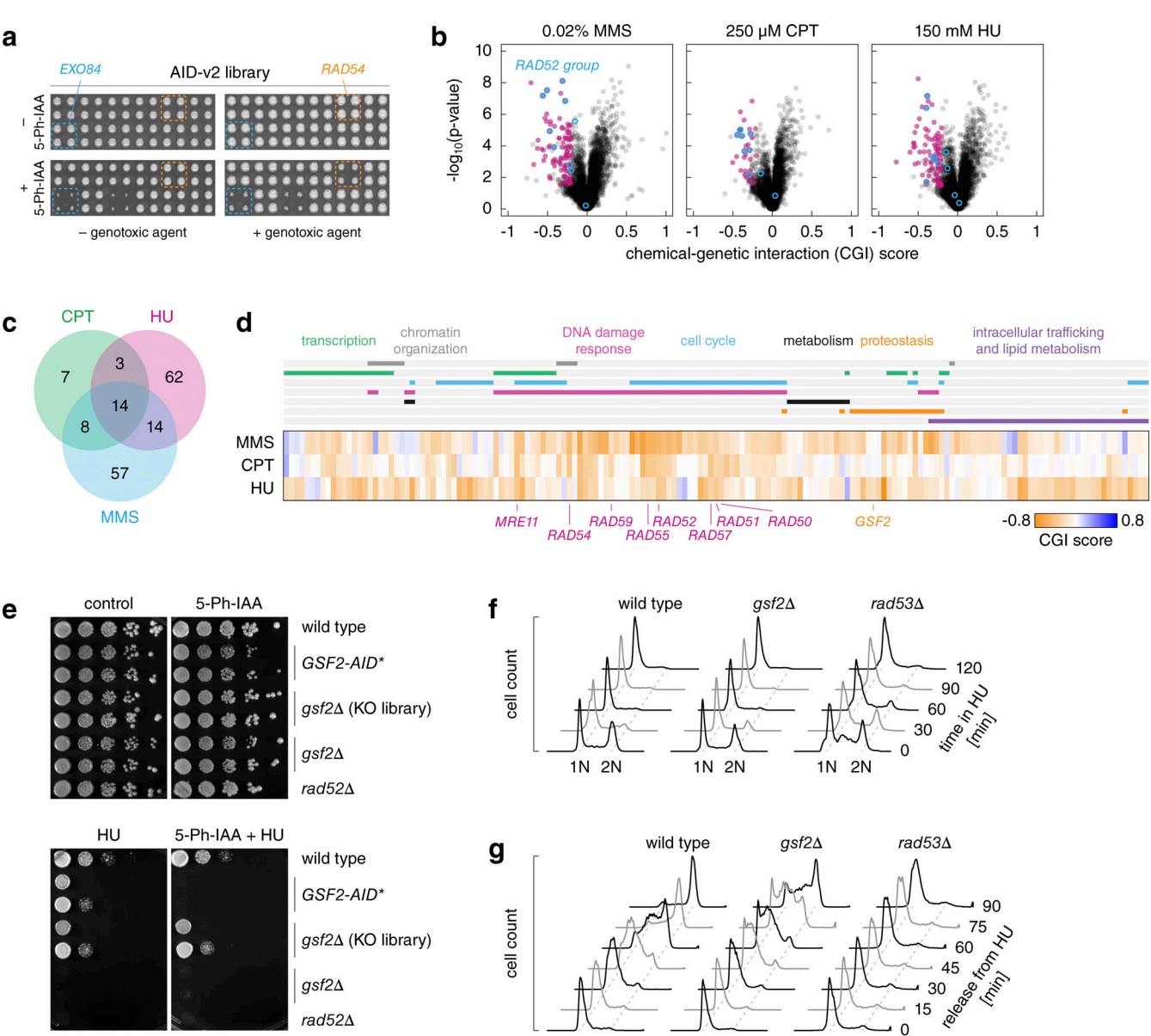

Figure 4. **Genome-wide screens for DNA damage response factors. (a)** Arrayed fitness screens. The AID-v2 library was grown on galactose media with or without MMS, CPT, or HU as GAs and 1 µM 5-Ph-IAA. **(b)** Volcano plots of CGI scores between the AID alleles and the indicated GAs. Negative chemical-genetic interactors with each GA (magenta, colony size [5-Ph-IAA + GA] ≤0.9, CGI score ≤0.2 and P value <0.05 in a two-sided *t* test, adjusted for multiple testing using the Benjamini-Hochberg method) and *RAD52* epistasis group genes (blue) are marked. **(c and d)** Venn diagram (c) and heatmap (d) of negative chemical-genetic interactors (magenta in b) with the three GAs. d, functional categories based on gene ontology terms (top, Materials and methods) and *RAD52* epistasis group genes (bottom) are indicated. **(e)** Sensitivity of *GSF2* mutants to HU. 10-fold serial dilutions of the indicated strains (GSF2-AID* strain from the AID-v2 library, *gsf2Δ* mutant from the haploid yeast knockout library and an independently constructed *gsf2Δ* strain) on galactose media with 5-Ph-IAA, HU, or both compounds. **(f and g)** Flow cytometry analysis of DNA content. Asynchronous cultures grown in galactose medium were treated with HU at time point zero (f). HU-arrested cultures were released into galactose medium without HU (g). G1 and G2 peaks are labeled 1N and 2N, respectively.

## Discussion

Our work establishes genome-wide libraries of conditional AID alleles in budding yeast. We demonstrate how these resources can be applied to uncover gene functions, thus adding to the toolkit for functional genomics in this organism (Botstein and Fink, 2011). While the extent of protein degradation in the AID libraries can be assessed in a high-throughput manner with mNG fluorescence, the optional nature of the mNG moiety expands the scope of potential applications. The AID-v2 library should be ideal for high-content fluorescence microscopy screens, where the mNG moiety could otherwise limit screen design. A similar resource was independently constructed and characterized by (Valenti et al., 2025). In the future, the libraries could be potentially improved with N-terminal tagging of ORFs that currently exhibit partial or no degradation of AID-tagged proteins or using multiple copies of the AID* tag to enhance protein degradation (Kubota et al., 2013; Nishimura and Kanemaki, 2014). Finally, it is noteworthy that a molecular glue

for a ubiquitin ligase located in the cytosol can be used to promote the degradation of the vast majority of yeast proteins, including integral membrane proteins. This suggests that targeted protein degraders such as proteolysis-targeting chimeras could in principle be designed for most proteins relevant in clinical or agricultural applications based on one or a few cytosolic ubiquitin ligases (Békés et al., 2022).

## Materials and methods

### Yeast strains and growth conditions

All yeast strains used in this work are listed in Table S6 and are derivatives of BY4741, Y8205, or Y7092 (Brachmann et al., 1998; Tong and Boone, 2007). All plasmids used in this work are listed in Table S7. Yeast genome manipulations (gene tagging and gene deletion) were performed using PCR targeting and lithium acetate transformation (Knop et al., 1999; Janke et al., 2004). High-throughput N-terminal and C-terminal tagging were performed with the SWAp-Tag approach using the N-SWAT (Weill et al., 2018) and C-SWAT (Meurer et al., 2018) libraries, respectively, as detailed below. Unless stated otherwise, yeast strains were grown at 30°C, and 5-Ph-IAA (30-003-10; BioAcademia) was used at 1 μM final concentration. All strains, plasmids, and libraries are available upon request.

### Construction of Halo libraries

To construct the Halo-ORF library, the donor plasmid pSD-N1-HaloTag-3myc was transformed into the donor strain yKBJ0201. The resulting strain yKBJ0203 was crossed with the N-SWAT library. In addition, to test the efficiency of tagging, the donor strain yMaM1205 was transformed with the donor plasmid pSD-N1-mNG and crossed with 13 N-SWAT strains for highly expressed proteins.

To construct the ORF-Halo library, the donor plasmid pRS41K-3myc-HaloTag was transformed into the donor strain yKBJ0200. The resulting strain yKBJ0202 was crossed with the C-SWAT library. In addition, to test the efficiency of tagging, the donor strain yKBJ0147 was transformed with the donor plasmid pRS41K-mNG and crossed with 13 N-SWAT strains for highly expressed proteins.

Crossing and subsequent tagging were performed by sequentially pinning the strains on appropriate media using a pinning robot (Rotor; Singer Instruments) in a 384-colony format according to the synthetic genetic array (SGA) procedure (Tong et al., 2001; Baryshnikova et al., 2010a; Meurer et al., 2018) as follows:

- N-SWAT and C-SWAT libraries were assembled in a 384-colony format on SC-Ura plates (6.7 g/liter yeast nitrogen base without amino acids [291940; BD Biosciences], 2 g/liter amino synthetic complete acid mix lacking uracil [SC-Ura], 20 g/liter glucose [108337; Merck], 20 g/liter agar [214010; BD Biosciences]), whereas donor strains with the appropriate donor plasmids were pinned onto yeast extract peptone dextrose (YPD) +G418 plates (10 g/liter yeast extract [212750; BD Biosciences], 20 g/liter peptone [211677; BD Biosciences], 20 g/liter glucose, 20 g/liter agar, 200 mg/liter G-418 [A291–25; Biochrom]), and grown for 1–2 days at 30°C;

- mating of N-SWAT/C-SWAT and donor strains on YPD plates, 1 day at 30°C;
- selection of diploids twice on SC(MSG)-Ura + G-418 plates (1.7 g/liter yeast nitrogen base without amino acids and ammonium sulfate [233520; BD Biosciences], 1 g/liter monosodium glutamic acid [MSG] [G1626; Sigma-Aldrich], 2 g/liter amino acid mix SC(MSG)-Ura [glutamic acid replaced by MSG], 200 mg/l G-418, 20 g/liter glucose, 20 g/liter agar), 1 day at 30°C;
- sporulation on SPO plates (20 g/liter potassium acetate [25059; Sigma-Aldrich], 20 g/liter agar), 5–7 days at 23°C;
- selection of haploids, step 1, on SC(MSG)-Leu/Arg/Lys/Ura + canavanine/thialysine plates (50 mg/liter canavanine [C1625; Sigma-Aldrich], 50 mg/liter thialysine [A2636; Sigma-Aldrich]), 2 days at 30°C;
- selection of haploids, step 2, on SC(MSG)-Leu/Arg/Lys/Ura + canavanine/thialysine/G-418 plates (50 mg/liter canavanine, 50 mg/liter thialysine, 200 mg/liter G-418), 2 days at 30°C;
- selection of haploids, step 3, on SC(MSG)-Leu/Arg/Lys/Ura + canavanine/thialysine/G-418/clonNAT plates (50 mg/liter canavanine, 50 mg/liter thialysine, 200 mg/liter G-418, 100 mg/liter clonNAT [5.0; Werner BioAgents]), 2 days at 30°C. This step was omitted for the cross used to assess tagging efficiency with N-SWAT strains;
- selection of haploids, step 4, on SC(MSG)-Leu/Arg/Lys/Ura + canavanine/thialysine/G-418/clonNAT/hygromycin plates (50 mg/liter canavanine, 50 mg/liter thialysine, 200 mg/liter G-418, 100 mg/liter clonNAT, 200 mg/liter hygromycin [ant-hg-5; Invivogen]), 2 days at 30°C. This step was omitted for the crosses used to assess tagging efficiency with N-SWAT and C-SWAT strains;
- induction of tagging twice on SC-Leu Raf/Gal plates (20 g/liter galactose [22020; Serva] and 20 g/liter raffinose [R0250; Sigma-Aldrich] instead of glucose), 2 days at 30°C;
- selection against the acceptor module twice on SC-Leu + 5-FOA plates (1 g/liter 5-FOA [PC4054; Apollo Scientific]), 2 days at 30°C; Finally, the resulting strains were pinned on SC-Leu+Ade plates (300 mg/liter adenine [A8626; Sigma-Aldrich]) and grown for 1 day at 30°C before proceeding with fluorescence measurements using flow cytometry and storing the libraries as glycerol stocks at –80°C.

### Construction of AID libraries

To construct the AID-v1 library, the donor plasmid pRS41K-mNG-AID*-3myc was transformed into the donor strains yEG222 (carrying the GAL1pr-OsTIR1(F74G)-natMX construct) and yMaM1205. To construct the AID-v2 library, the donor plasmid pRS41K-AID*-3myc was transformed into the donor strain yEG222. The resulting strains were crossed with the C-SWAT library. In addition, to test the efficiency of tagging, the donor strain yEG222 was transformed with the donor plasmid pRS41K-mCherry-mNG and crossed with 13 C-SWAT strains for highly expressed proteins.

Crossing and subsequent tagging were performed by sequentially pinning the strains on appropriate media in a 384-colony format using the same procedure as for the Halo libraries with the following changes:

- selection of diploids twice on SC(MSG)-Ura + G-418/clonNAT plates, 1 day at 30°C;
- selection of haploids, step 1, on SC(MSG)-Leu/Arg/Lys/Ura + canavanine/ thialysine/G-418 plates, 2 days at 30°C;
- selection of haploids, step 2, on SC(MSG)-Leu/Arg/Lys/Ura + canavanine/ thialysine/G-418/clonNAT plates, 2 days at 30°C. This step was omitted for the cross used to assess tagging efficiency with the pRS41K-mCherry-mNG donor plasmid;
- induction of tagging twice on SC-Leu Raf/Gal plates (20 g/liter galactose [22020; Serva] and 20 g/liter raffinose [R0250; Sigma-Aldrich] instead of glucose), 2 days at 30°C;
- selection against the acceptor module twice on SC-Leu + 5-FOA plates (1 g/liter 5-FOA [PC4054; Apollo Scientific]), 2 days at 30°C;
- selection of successful tagging events with reconstitution of the hygromycin selection marker (after induction of tagging and selection against the acceptor module) twice on SC(MSG)-Leu + hygromycin/clonNAT plates (400 mg/liter hygromycin, 100 mg/liter clonNAT), 2 days at 30°C.

The same procedure but without clonNAT was used to construct the AID-v1 library without the GAL1pr-OsTIR1(F74G)-natMX construct.

### Competition assay

Using a pinning robot, strains from the Halo-ORF and ORF-Halo libraries were co-inoculated in 96-well plates (83.3924.005; Sarstedt) with 200 µl of SC-Leu+Ade medium per well. Each well received the two strains for a given ORF, one N-terminally tagged and carrying a cytosolic mNG marker and another C-terminally tagged and carrying a cytosolic mCherry marker. Three control wells ([1] non-fluorescent strain yMaM1205, [2] mCherry-positive donor yKBJ0202, [3] mNG-positive donor yKBJ0203, Table S6) served to set the gates for mCherry+ and mNG+ cells per plate. The plates were sealed with air-permeable seals (4ti-0517; 4Titude) and the co-cultures were grown overnight at 30°C to saturation.

Saturated co-cultures were diluted (5 µl of co-culture into 195 µl of SC-Leu+Ade medium) and grown for 4 h at 30°C in sealed plates before single-cell fluorescence measurements with flow cytometry corresponding to day 0. After the measurements, the plates with the remaining 175 µl of co-culture per well were sealed and incubated for 20 h at 30°C. The procedure starting with dilution of co-cultures was then repeated three times (flow cytometry analysis corresponding to days 1–3, 24-h interval between measurements).

Triple competition experiments for 45 ORFs selected at random were conducted using the same procedure. Single clones of Halo-ORF and ORF-Halo strains were validated by immunoblotting of whole-cell extracts with anti-myc antibodies (mouse monoclonal, clone 9E10). Matching Halo-ORF and ORF-Halo strains were grown in each well together with a wild type strain lacking fluorescent markers (yMaM1205, Table S6).

### Flow cytometry

Strains were inoculated in 96-well plates (83.3924.005; Sarstedt) with 150–200 µl of SC-Leu+Ade medium per well and grown overnight at 30°C to saturation. Saturated cultures were diluted into fresh medium and grown at 30°C for 6 h to 6–8 × 10⁶ cells/ ml. Assessment of tagging efficiency during the construction of Halo libraries and the high-throughput competition assays were performed with strains grown in SC-Leu+Ade medium.

Single-cell fluorescence intensities were measured on an LSRFortessa SORP flow cytometer (BD Biosciences) using the BD FACSDiva v9.0.1 acquisition software, a 561-nm laser for mCherry excitation, a 600-nm long pass mirror and a 610/20-nm band pass filter for mCherry detection, a 488-nm laser for mNG excitation, a 505-nm long pass mirror, and a 530/30-nm band pass filter for mNG detection and a high throughput sampler with the following setting: standard mode, sample flow rate of 0.5 µl/s, sample volume of 25 µl, mixing volume of 100 µl, mixing speed of 100 µl/s, number of mixes equal to 3 and wash volume of 800 µl. 10,000 cells were measured per strain. Measurements were gated for single cells with mCherry or mNG fluorescence intensities above the maximum intensity of a non-fluorescent control strain (mCherry+ or mNG+ cells). In the triple competition assay, measurements were additionally gated for cells lacking a fluorescent marker (wild type cells) (Fig. S6).

### Analysis of competition experiments

Data analysis and visualization were performed using custom scripts in R (R Core Team, 2020). Ratios of mNG+/mCherry+ cells were calculated for each well and time point from single-cell fluorescence measurements obtained with flow cytometry. For each well, cell count ratios were normalized to time point zero. The relative fitness of the two strains in a well was then determined as the slope α of a linear fit to ln(normalized cell count ratio) over time, with the intercept set to zero. Triple competition experiments were analyzed following the same procedure for three cell count ratios: mNG+/mCherry+ cells, mNG+/wild type cells, and mCherry+/wild type cells.

Using optical density measurements at 600 nm of 15 wells, we estimated an average doubling time of 4.5 ± 0.1 h (mean ± SD, $n$ = 15 random ORFs) in the competition assay. The average doubling time allowed translating the relative fitness α into a difference in doubling time under the assumption that, for a given protein, tagging of only one of the termini affects fitness and that $α ≠ 0$ reflects impaired fitness, not improved fitness, of one of the competing strains. Based on this and on estimates of reproducibility and technical noise in the competition assay (Fig. S1, b and c), further analysis of differential fitness relied on two thresholds of abs(α) = 0.175 and 0.611 corresponding to 5% and 20%, respectively, longer doubling time for one of the competing strains.

The annotation of yeast genes as essential or non-essential under standard laboratory growth conditions (Winzeler et al., 1999; Giaever et al., 2002; Després et al., 2020) and a proteome-wide localization dataset of C-terminally GFP-tagged proteins (Huh et al., 2003) were used to stratify the genome-wide distribution of relative fitness (Fig. 1, c and d). Proteins assigned to multiple localization classes were counted in each class.

The relationship between differential localization and differential fitness of N- and C-terminally tagged proteins (Fig. 1 e)

was examined using proteome-wide localization datasets of N-terminally GFP-tagged proteins expressed from endogenous promoters and of C-terminally GFP-tagged proteins (Huh et al., 2003; Weill et al., 2018). The localization classes, namely, *missing*, *ambiguous*, and *below threshold* were excluded from the analysis and only proteins with annotated localization in both datasets were considered. N- and C-terminally tagged variants assigned to multiple localization classes were considered to have the same localization if they shared at least one localization assignment (Weill et al., 2018).

The relationship between terminal localization signals and differential fitness of N- and C-terminally tagged strains was examined using a list of 998 validated or predicted localization signals obtained through literature curation or retrieved from the UniProt database (Bateman et al., 2023) (Table S2). The list consists of proteins with a glycosylphosphatidylinositol (GPI) anchor (de Groot et al., 2003; Gíslason et al., 2021), tail-anchored proteins (Beilharz et al., 2003; Burri and Lithgow, 2004; Schuldiner et al., 2008; Simocková et al., 2008; Lewis and Pelham, 2002), proteins with a cysteine-rich transmembrane module (Venancio and Aravind, 2010; Giolito et al., 2024; Zvonarev et al., 2023), acylated proteins (UniProt as of February 19, 2024, search for proteins with lipidation, including N-myristoylation, palmitoylation, GPI-anchor addition and prenylation), proteins with a PTS1 peroxisomal targeting signal (Nötzel et al., 2016) (UniProt as of April 24, 2024, search for proteins with motif: microbody targeting signal, C-terminal peroxisome targeting signal [PTS1], peroxisome targeting signal [PTS1], SKL peroxisome targeting motif, peroxisomal target signal 1 [PTS1], or peroxisomal targeting signal type 1), proteins with a signal peptide or an MTS (Somashekara and Muniyappa, 2022; Yofe et al., 2016; Weill et al., 2018), proteins with nuclear localization signals (Hahn et al., 2008; Lee et al., 2023; Bordonné, 2000; Somashekara and Muniyappa, 2022), and proteins with HDEL, KKXX or KXKXX motifs identified by pattern match (Yan et al., 2005) or retrieved from UniProt (UniProt as of April 24, 2024, search for proteins with motif: prevents secretion from ER or di-lysine motif).

**Colony fluorescence measurements and analysis**
The AID-v1 library was assembled in 1,536-colony format on SC(MSG)-Leu+Ade Raf/Gal + hygromycin plates using a pinning robot (Rotor; Singer Instruments). For each ORF, six technical replicates of the *OsTIR1*⁺ strain, three technical replicates of the *OsTIR1*⁻ strain, three technical replicates of a non-fluorescent strain for background correction (AID-v2_HIS4, Table S6) and four technical replicates of an mNG⁺ reference strain (AID-v1_OsTIR1⁻_PDC1, Table S6) were arranged in a 4 × 4 group (Fig. 2 c).

Assembled colony arrays were pinned onto SC(MSG)-Leu+Ade Raf/Gal + hygromycin + 5-Ph-IAA plates, grown for 24 h at 30°C and their fluorescence was measured with a multimode microplate reader equipped with monochromators for precise selection of excitation and emission wavelengths (Spark; Tecan) and a custom temperature-controlled incubation chamber. mNG fluorescence intensities were measured with 506/5-nm excitation, 524/5-nm emission, 40-µs integration time, and two detector gains for optimal detection of proteins across the full range of

expression levels in the experiment. In addition, the plates were photographed with a colony imager (Phenobooth; Singer Instruments) to assess strain fitness based on colony size, as detailed in the next section.

Data analysis and visualization were performed using custom scripts in R (R Core Team, 2020). Measurements at the two detector gains were interpolated and consolidated into a single value per colony. For each ORF, measurements were normalized by the average of the reference colonies in the 4 × 4 group, to correct for spatial and plate effects. Sample measurements were further corrected for background fluorescence (mNG-bkg), by subtraction of the mean of the non-fluorescent colonies in the 4 × 4 group, or expressed in units of background fluorescence (mNG/bkg), by dividing by the mean of the non-fluorescent colonies in the group, and summarized by the mean and SD of *OsTIR1*⁺ or *OsTIR1*⁻ replicates.

Background-corrected intensities of *OsTIR1*⁺ or *OsTIR1*⁻ strains were compared with a two-sided $t$ test and P values were adjusted for multiple testing using the Benjamini-Hochberg method. mNG-AID*-3myc-tagged proteins were subsequently classified based on their expression levels and the extent of *OsTIR1*-dependent degradation as follows:

- not detected: mNG/bkg(*OsTIR1*⁻) ≤ 1.2;
- degraded: mNG/bkg(*OsTIR1*⁻) > 1.2, mNG/bkg(*OsTIR1*⁺) ≤ 1.2, mNG-bkg(*OsTIR1*⁺)/mNG-bkg(*OsTIR1*⁻) < 0.5, and P value <0.05;
- partially degraded: mNG/bkg(*OsTIR1*⁻) > 1.2, mNG/bkg(*OsTIR1*⁺) > 1.2, mNG-bkg(*OsTIR1*⁺)/ mNG-bkg(*OsTIR1*⁻) < 0.5, and P value <0.05);
- not affected: mNG/bkg(*OsTIR1*⁻) > 1.2, mNG-bkg(*OsTIR1*⁺)/ mNG-bkg(*OsTIR1*⁻) ≥ 0.5 or mNG/bkg(*OsTIR1*⁻) > 1.2, P value ≥0.05.

Background-corrected intensities of *OsTIR1*⁻ strains were compared to measurements of absolute protein levels in molecules per cell (Lawless et al., 2016) and fluorescence measurements of a library of mNG-tagged ORFs (mNG-II library) (Meurer et al., 2018) (Fig. S2, c and d). Furthermore, proteins detected in the *OsTIR1*⁻ background were split into five abundance bins of equal width based on background-corrected mNG intensities (Fig. S2 f). The annotation of yeast genes as essential or non-essential under standard laboratory growth conditions (Winzeler et al., 1999; Giaever et al., 2002; Després et al., 2020), a proteome-wide localization dataset of C-terminally GFP-tagged proteins (Huh et al., 2003), a dataset of membrane protein topology (Kim et al., 2006), and annotations of essential genes as core or conditional (van Leeuwen et al., 2020; Peter et al., 2018) were used to further stratify the categorized (according to expression, extent of *OsTIR1*-dependent degradation, or *OsTIR1*-dependent fitness impairment) proteins (Fig. 2 e; Fig. 3 d; Fig. S3, c and d; and Fig. S4 e).

**Arrayed fitness screens and analysis**
The AID-v2 library was assembled in 1,536-colony format on SC(MSG)-Leu + hygromycin/clonNAT plates using a pinning robot (Rotor; Singer Instruments), with four technical replicates per strain placed next to each other in a 2 × 2 group. After 24 h of growth, the colony arrays were pinned on different media with

1 μM 5-Ph-IAA and GAs: control (YP Raf/Gal + hygromycin/clonNat), 5-Ph-IAA (YP Raf/Gal + hygromycin/clonNat/5-Ph-IAA), GA (YP Raf/Gal + hygromycin/clonNat/GA), and 5-Ph-IAA + GA (YP Raf/Gal + hygromycin/clonNat/5-Ph-IAA/GA) plates, where GA stands for CPT (250 μM; 7689-03-4; Sigma-Aldrich), HU (150 mM; 127-07-1; Sigma-Aldrich), or MMS (0.02% vol/vol; 66-27-3; Sigma-Aldrich).

Colony arrays were grown at 30°C and photographed after 24 and 48 h with a colony imager (Phenobooth; Singer Instruments). Subsequent data analysis and visualization were performed using custom scripts in R (R Core Team, 2020). The gitter package was used to segment the photographs and determine colony sizes (Wagih and Parts, 2014). The SGA tools package was used to correct for spatial and plate effects (Wagih et al., 2013): four outer rows and columns of dummy colonies on each plate were removed; spatial correction was performed without the row and column corrections; corrected measurements were normalized by the median of each plate (median calculated using only colonies between the 40th and 80th percentiles due to the high number of strains with impaired fitness). For each condition, the measurements were then normalized by the median of that condition (median of the data between the 40th and 80th percentiles) and summarized by the mean and SD for each strain.

A similar procedure was used to determine the colony sizes in the AID-v1 library grown according to Fig. 2 c. Next, the impact of the *GAL1-OsTIR1(F74G)* construct was estimated at 0.93 as the colony size of an *OsTIR1*⁺ strain relative to the corresponding *OsTIR1*⁻ strain, averaged across non-essential ORFs (Fig. S3 a). Colony size measurements of *OsTIR1*⁺ strains were then corrected for the fitness impact of OsTir1 expression using the multiplicative model (Baryshnikova et al., 2010b), whereby the fitness impacts of OsTir1 expression and degradation of the AID-tagged protein are independent: size(*OsTIR1*⁺, corrected) = size(*OsTIR1*⁺)/0.93.

To identify strains with impaired growth upon degradation of the AID-tagged protein, corrected colony sizes of *OsTIR1*⁺ and normalized colony sizes of *OsTIR1*⁻ strains in the AID-v1 library or normalized colony sizes of AID-v2 strains on control and 5-Ph-IAA plates were compared with a two-sided $t$ test and P values were adjusted for multiple testing using the Benjamini-Hochberg method. Strains with impaired fitness were defined as follows:

- AID-v1 library: size(*OsTIR1*⁺)/size(*OsTIR1*⁻) < 0.8 and P value <0.05 (Fig. 3 a);
- AID-v2 library: size(5-Ph-IAA)/size(control) < 0.8 and P value <0.05 (Fig. S4 b).

CGI scores were calculated for each ORF using the normalized median-centered colony sizes and the multiplicative model (Baryshnikova et al., 2010b) as follows:

CGI score = size(5-Ph-IAA, GA) – size(5-Ph-IAA) × size(GA).

Finally, colony sizes in the double perturbation condition (5-Ph-IAA, GA) were compared against the product of sizes from the single perturbation (5-Ph-IAA and GA) with a two-sided $t$ test and P values were adjusted for multiple testing using the Benjamini-Hochberg method. Significant negative CGIs were defined as follows: size(5-Ph-IAA, GA) ≤ 0.9, CGI score ≤ –0.2, and P value <0.05. The identified genes were mapped to gene ontology (GO) terms using the GO slim mapper in the Saccharomyces genome database with GO version 2024-01 (Wong et al., 2023) and grouped into broader functional categories (Fig. 4 d and Fig. S5 c) as follows:

- transcription GO terms: transcription by RNA polymerase I, transcription by RNA polymerase II, mRNA processing, rRNA processing, and RNA catabolic process;
- chromatin organization GO terms: chromatin organization;
- DNA damage response GO terms: DNA repair, DNA recombination, and DNA damage response;
- cell cycle GO terms: meiotic cell cycle, mitotic cell cycle, regulation of cell cycle, DNA replication, sporulation, and chromosome segregation;
- metabolism GO terms: amino acid metabolic process, vitamin metabolic process, generation of precursor metabolites and energy, carbohydrate metabolic process, and nucleobase-containing small molecule metabolic process;
- proteostasis GO terms: protein glycosylation, protein folding, protein modification by small protein conjugation or removal, and proteolysis involved in protein catabolic process;
- intracellular trafficking and lipid metabolism GO terms: endosomal transport, transmembrane transport, vacuole organization, endocytosis, golgi vesicle transport, membrane fusion, vesicle organization, lipid metabolic process, and protein targeting.

## Yeast spot assays
Strains were grown overnight in YPD medium at 30°C. 10-fold serial dilutions, starting with 0.5 OD$_{600\,nm}$, were spotted onto agar plates using a replica plater (R2383; Sigma-Aldrich) and incubated at 30°C. The plates were photographed (ChemiDoc Touch imaging system; Bio-Rad) after 2–4 days of incubation.

## Time courses of inducible protein degradation
Strains were grown overnight in YP Raf medium at 30°C, diluted to 0.2 OD$_{600\,nm}$, and incubated for 3 h, followed by the addition of galactose to 2% (wt/vol) to induce expression of *OsTIR1*. Time courses were initiated 90 min later with the addition of 5-Ph-IAA to 1 μM final concentration and 1.0 OD$_{600\,nm}$ samples were collected at 0, 10, 30, and 60 min time points.

Whole-cell extracts were prepared by alkaline lysis followed by precipitation with trichloroacetic acid (Knop et al., 1999) and separated by SDS-PAGE, followed by immunoblotting against the myc tag and against the loading control Pgk1 with mouse anti-myc (1:1,000 dilution, clone 9E10; Institute of Molecular Biology [IMB] Protein Production core facility), mouse anti-Pgk1 (1:10,000 dilution, 459250; Thermo Fisher Scientific), and goat HRP-conjugated anti-mouse (1:3,000 dilution, 1705047; Bio-Rad) antibodies. Nitrocellulose 0.2 μm membranes (1704271; Bio-Rad) were developed using a chemiluminescent horseradish peroxidase substrate (SuperSignal West Pico PLUS, 34580; Thermo Fisher Scientific) and imaged using a ChemiDoc Touch imaging system (Bio-Rad) with auto-exposure settings and 4 × 4 binning.

### Cell cycle profiling

Overnight cultures in YP Raf medium were diluted to 0.2 $OD_{600\,nm}$ and incubated at 30°C. After 3 h, galactose was added to 2% (wt/vol) to induce the expression of *OsTIR1*. Time courses were initiated 90 min later with the addition of HU to 150 mM final concentration and 1 ml samples were collected every 15 min for 2 h. At the 2-h time point, cultures were pelleted by centrifugation, washed three times with prewarmed YP Raf/Gal, and released into YP Raf/Gal medium at 25°C. To follow the release from HU, samples were collected every 15 min for 90 min.

At each time point, cells were harvested by centrifugation and fixed in 70% ethanol overnight at 4°C. Fixed cells were pelleted by centrifugation at 17,000 *g* for 5 min, washed once with 800 µl of 50 mM sodium citrate pH 7.4 buffer, resuspended in 500 µl of 50 mM sodium citrate pH 7.4 buffer, treated with 0.25 mg/ml RNase A (10753721; Thermo Fisher Scientific) for 3 h at 37°C, followed by treatment with 1 mg/ml Proteinase K (M3037.0005; Genaxxon) for 2 h at 50°C. Treated samples were stained by the addition of SYTOX Green (1076273; Thermo Fisher Scientific) to 4 µM final concentration. Cell cycle profiles were measured with a flow cytometer (LSRFortessa; BD Biosciences with BD FACSDiva software v9.7), with 10,000 recorded events per sample. Data analysis was performed with FlowJo v10.10 (BD Biosciences).

### Online supplemental material

Fig. S1 shows the fitness impact of N- versus C-terminal tagging. Fig. S2 shows the construction and characterization of genome-wide AID libraries. Fig. S3 shows characterization of the AID-v1 library. Fig. S4 shows the characterization of the AID-v2 library. Fig. S5 shows genome-wide screens for DNA damage response factors. Fig. S6 shows a representative gating strategy used in flow cytometry experiments. Table S1 shows the results of the genome-wide competition screen. Table S2 shows terminal localization signals in the yeast proteome. Table S3 shows the characterization of the AID-v1 library. Table S4 shows the results of the genome-wide screens for DNA damage response factors. Table S5 shows gene set enrichment analysis of resistance factors unique to each GA. Table S6 shows yeast strains used in this study. Table S7 shows the plasmids used in this study.

### Data availability

The results of the genome-wide competition screen, characterization of the AID-v1 and AID-v2 library, and the genome-wide screens for DNA damage response factors are provided as Tables S1, S3, and S4, respectively. The pipelines for downstream analysis and data visualization, together with the necessary input data, are available at https://github.com/khmelinskii-lab/AID_libraries.

## Acknowledgments

We thank Helle Ulrich (IMB, Mainz, Germany) for reagents. We thank the IMB Flow Cytometry core facility for the use of their instruments and Stefanie Möckel for help with the experimental setup of the competition screen and initial data analysis, and the IMB Protein Production core facility for the mouse anti-myc antibody.

Funding of the German Research Foundation supported the BD LSRFortessa Special Order Research Product (P#210253511). The IMB Media Lab and the IMB Protein Production facility are gratefully acknowledged for support with growth media and reagents. This work was supported by the European Research Council (ERC-2017-STG#759427 to A. Khmelinskii) and by the Institute for Quantitative and Computational Biosciences.

Author contributions: E. Gameiro: Formal analysis, Investigation, Writing - review & editing, K.A. Juárez-Núñez: Data curation, Formal analysis, Investigation, Methodology, Software, Validation, Visualization, Writing - review & editing, J.J. Fung: Data curation, Formal analysis, Software, S. Shankar: Software, B. Luke: Conceptualization, Funding acquisition, Methodology, Resources, Supervision, Writing - original draft, Writing - review & editing, A. Khmelinskii: Conceptualization, Funding acquisition, Supervision, Writing - original draft, Writing - review & editing.

Disclosures: The authors declare no competing interests exist.

Submitted: 3 September 2024

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

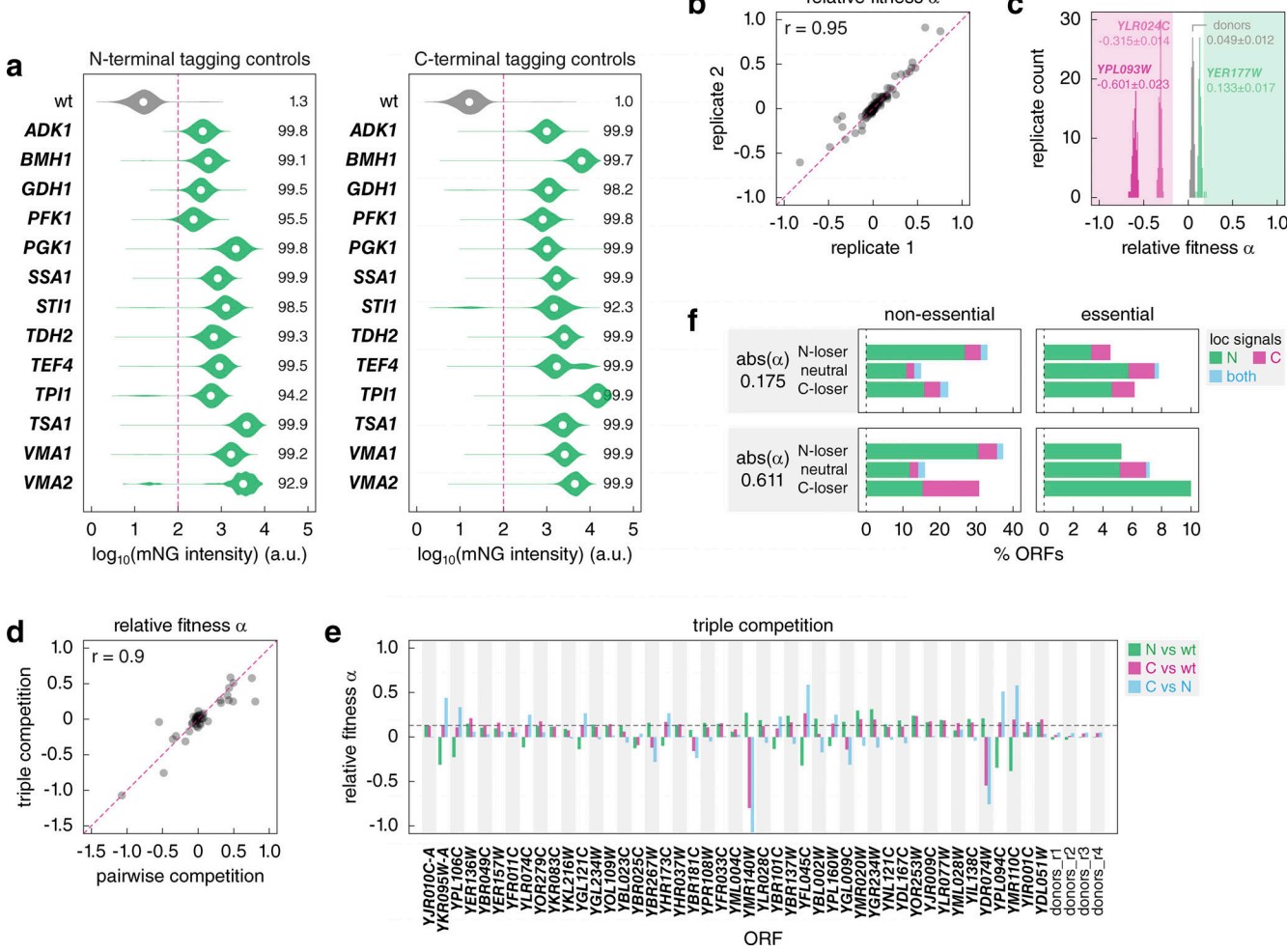

Figure S1. **Fitness impact of N- versus C-terminal tagging. (a)** Evaluation of the SWAT tagging efficiency during construction of the Halo-ORF and ORF-Halo libraries using donors with mNG as the tag. Tagging was performed with 13 N-SWAT (left) or C-SWAT (right) strains for the indicated ORFs. Distributions of single-cell mNG fluorescence intensities measured with flow cytometry (10,000 cells per strain). The percentage of cells with fluorescence above the background (fluorescence of a wild type [wt] strain, dashed line) is indicated. **(b)** Reproducibility of relative fitness estimation with the competition assay. Relative fitness of two replicates for 92 ORFs, determined as in Fig. 1 a. r, Pearson correlation coefficient. **(c)** Evaluation of technical noise in the competition assay. Distributions of relative fitness determined as in Fig. 1 a for the indicated ORFs and the pair of SWAT donor strains (no tagged ORF), 92 technical replicates per strain pair. Mean ± SD relative fitness is indicated. **(d and e)** Triple competition of 45 randomly selected pairs of Halo-ORF and ORF-Halo strains against a wild type strain lacking fluorescent markers. Relative fitness of Halo-tagged strains determined according to Fig. 1 a and in the triple competition assay (d, Materials and methods). Pairwise relative fitness of the three strains for each ORF in the triple competition assay (e, Materials and methods). Median relative fitness (0.13) of N- and C-terminal variants against wild type (dashed line, e). **(f)** Proteins with localization signals at the N-terminus, C-terminus, or both protein termini, stratified by gene essentiality and differential fitness according to Fig. 1 b. Analysis based on 997 validated or predicted localization signals: 611, 188, and 99 proteins with signals at the N-terminus, the C-terminus, or at both termini, respectively (Materials and methods, Table S2). N-loser, neutral, and C-loser ORFs are defined at two relative fitness thresholds.

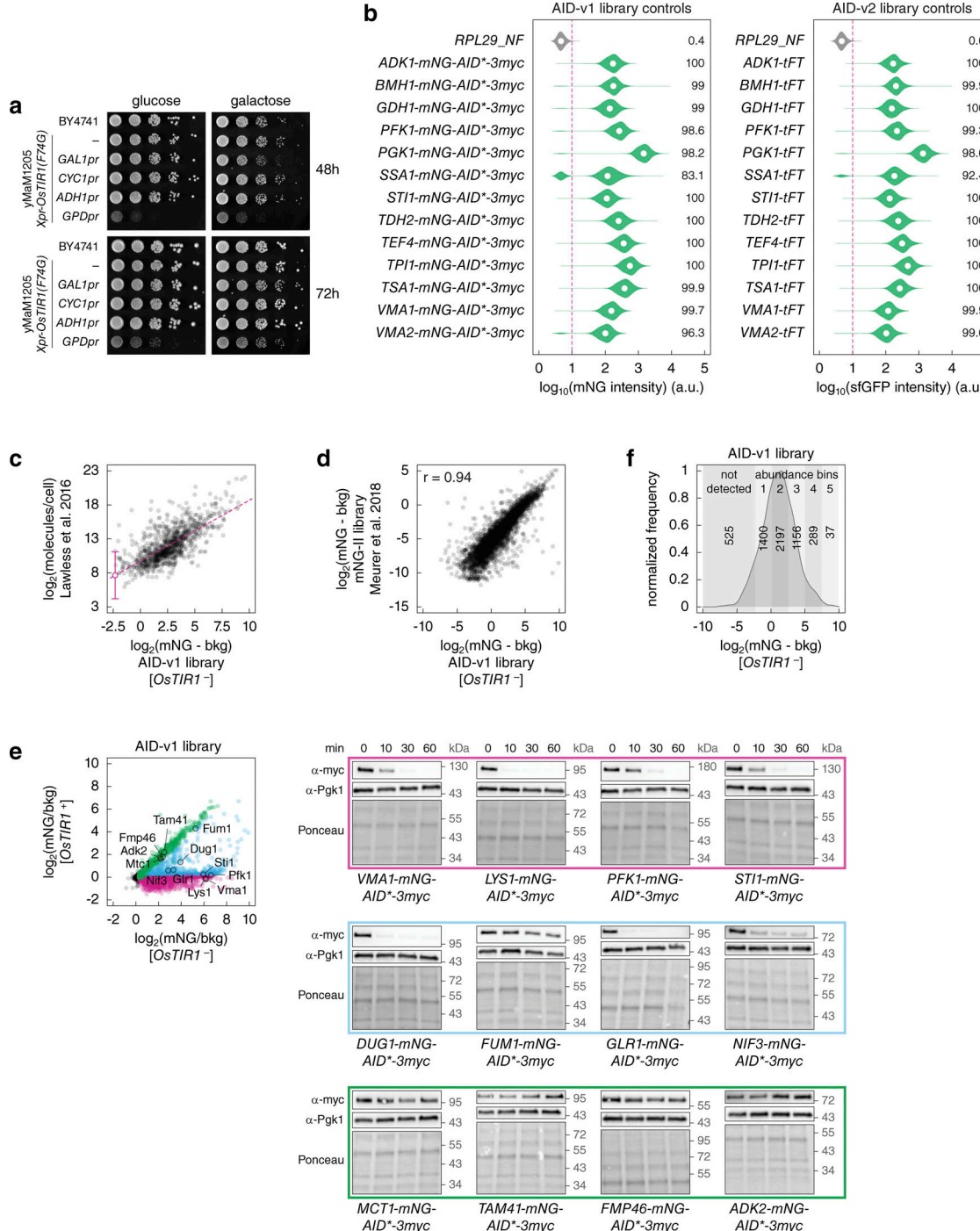

Figure S2. **Construction and characterization of genome-wide AID libraries. (a)** Fitness impact of *OsTIR1* expression from different promoters. 10-fold serial dilutions of yeast strains on the indicated media without 5-Ph-IAA. **(b)** Evaluation of the SWAT tagging efficiency during construction of the AID libraries. For the AID-v2 library, control tagging was performed with 13 C-SWAT strains for the indicated ORFs and a donor with the mCherry-sfGFP timer (tFT) as a tag. Distributions of single-cell mNG fluorescence intensities measured with flow cytometry (10,000 cells per strain). The percentage of cells with fluorescence above background (fluorescence of a wild type strain, dashed line) is indicated. **(c)** Correlation between absolute protein abundance (Lawless et al., 2016) and relative protein expression levels measured with the AID-v1 library lacking *OsTIR1*. Fluorescence intensities of colonies were corrected for background fluorescence. Linear fit (dashed line) and estimate of the detection limit of the colony assay (200 molecules/cell) with the 95% confidence interval (18 to 2,187 molecules/cell). **(d)** Comparison of relative protein expression levels measured with the AID-v1 library lacking *OsTIR1* and with the mNG-II library (ORF-mNG strains) (Meurer et al., 2018). Fluorescence intensities of colonies were corrected for background fluorescence. r, Pearson correlation coefficient. **(e)** Degradation of AID-tagged proteins after addition of 5-Ph-IAA. Whole-cell extracts of *OsTIR1*+ strains for the indicated proteins (AID-v1 library, left) were separated by SDS-PAGE, followed by immunoblotting with antibodies against the myc tag and Pgk1 as the loading control (right). **(f)** Distribution of background-corrected mNG fluorescence intensities in the AID-v1 library, *OsTIR1*− strains, determined according to Fig. 2 c. Number of proteins in the not detected group (mNG/bkg(OsTIR1−) ≤ 1.2) and in five abundance bins used in Fig. 2 e; Fig. 3 d; Fig. S3 b; and Fig. S4 e is indicated. Source data are available for this figure: SourceData FS2.

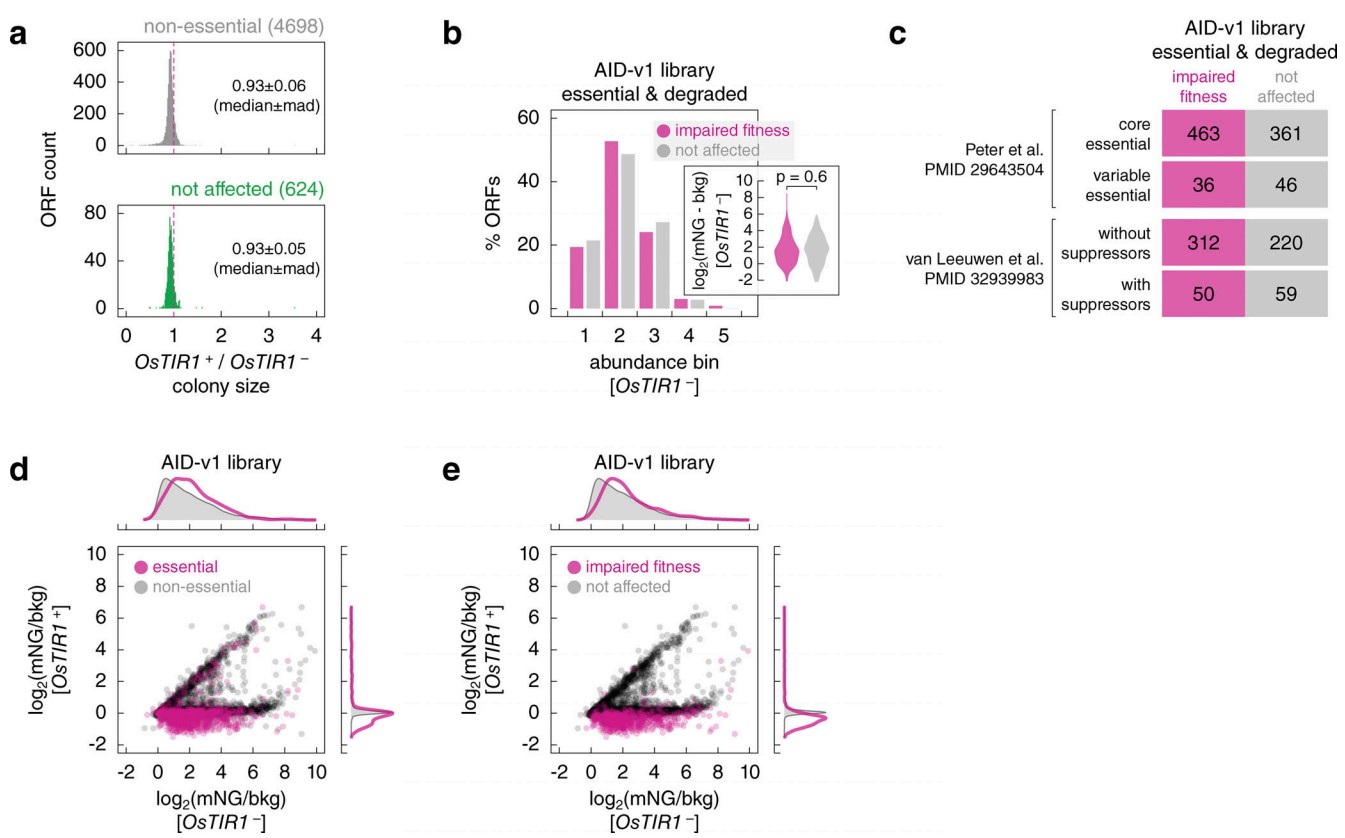

Figure S3. **Characterization of the AID-v1 library. (a)** Fitness impact of the *GAL1pr-OsTIR1(F74G)* expression construct. For each ORF, the fitness impact was estimated as the colony size of the *OsTIR1⁺* strain relative to the *OsTIR1⁻* strain in the AID-v1 library (*OsTIR1⁺*/*OsTIR1⁻*), determined according to Fig. 2 c. Distributions of relative colony sizes for two sets of ORFs (4,698 non-essential ORFs or 624 not affected proteins in Fig. 2 d). Median ± mad (median absolute deviation) relative fitness is indicated. **(b)** Expression levels of essential and degraded proteins in the AID-v1 library (Fig. 2 c), according to fitness of *OsTIR1⁺* strains. Protein abundance (mNG signal corrected for background fluorescence, mNG-bkg) in the *OsTIR1⁻* background; abundance bins defined in Fig. S2 f; P value in a Mann–Whitney U test. **(c)** Number of *OsTIR1⁺* strains in the AID-v1 library with essential and degraded proteins, stratified by essential gene type: core or variable according to Peter et al. (2018), with or without suppressors according to van Leeuwen et al. (2020). **(d and e)** Protein levels (mNG signal in units of background fluorescence, mNG/bkg) in the AID-v1 library determined according to Fig. 2 c. Essential and non-essential ORFs (d) and strains with normal or impaired fitness based on colony size measurements (e) are marked. Top and right, normalized frequency distributions of protein levels in each category.

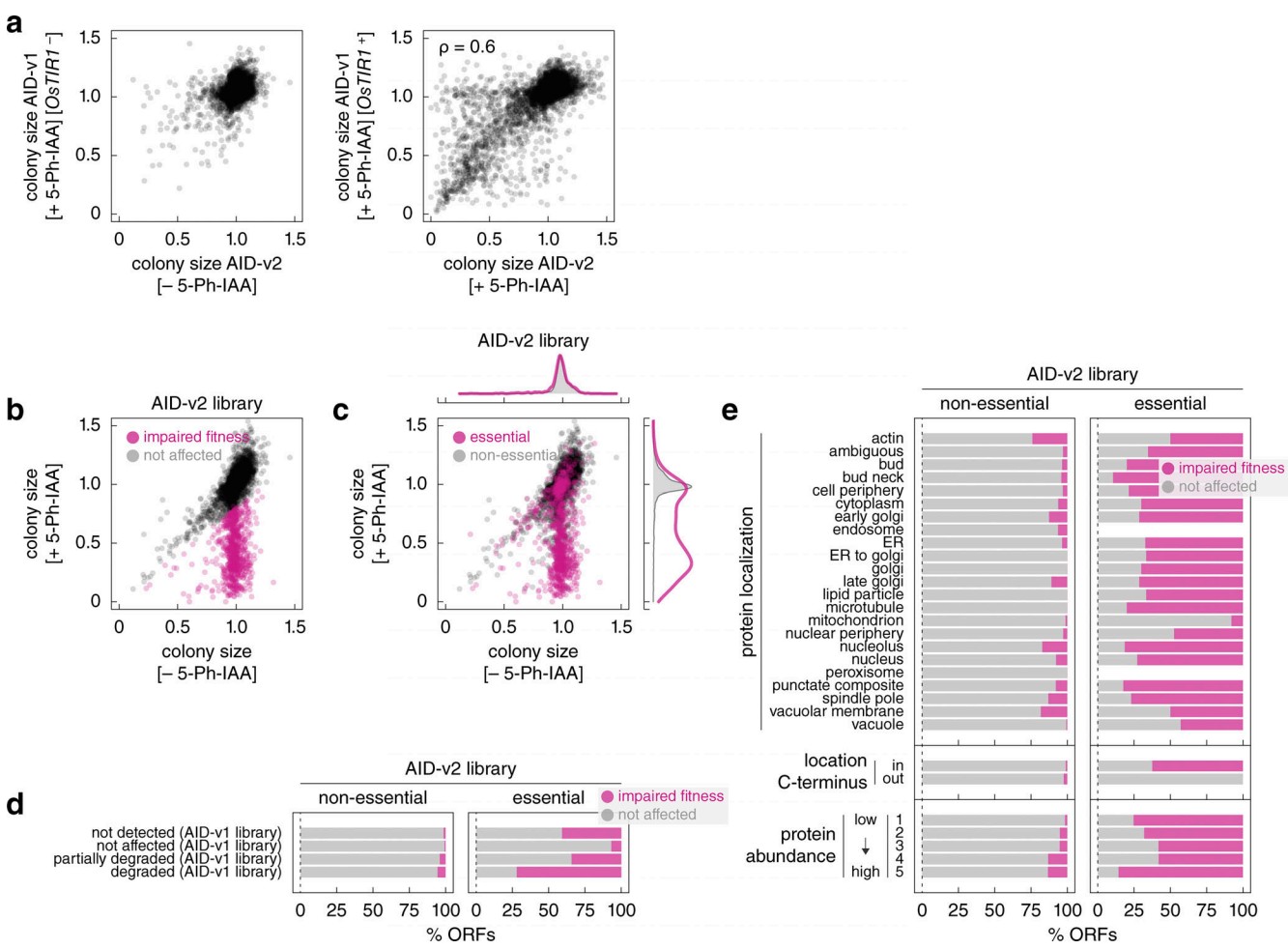

Figure S4.   **Characterization of the AID-v2 library. (a)** Comparison of normalized colony sizes in the AID-v1 and AID-v2 libraries grown according to Fig. 2 c and Fig. 4 a, respectively. ρ, spearman correlation coefficient. **(b and c)** Normalized colony sizes in the AID-v2 library grown according to Fig. 4 a on galactose medium with or without 1 μM 5-Ph-IAA. c, essential and non-essential ORFs are marked; top and right, normalized frequency distributions of normalized colony sizes in each category. **(d and e)** Frequency of 5-Ph-IAA–dependent fitness defects in the AID-v2 library according to the extent of *OsTIR1*-dependent degradation phenotypes in the AID-v1 library (d) or according to protein localization, location of the C-terminus (cytosol – in, lumen of organelles or extracellular – out) or protein abundance estimates from the AID1-v1 library (e).

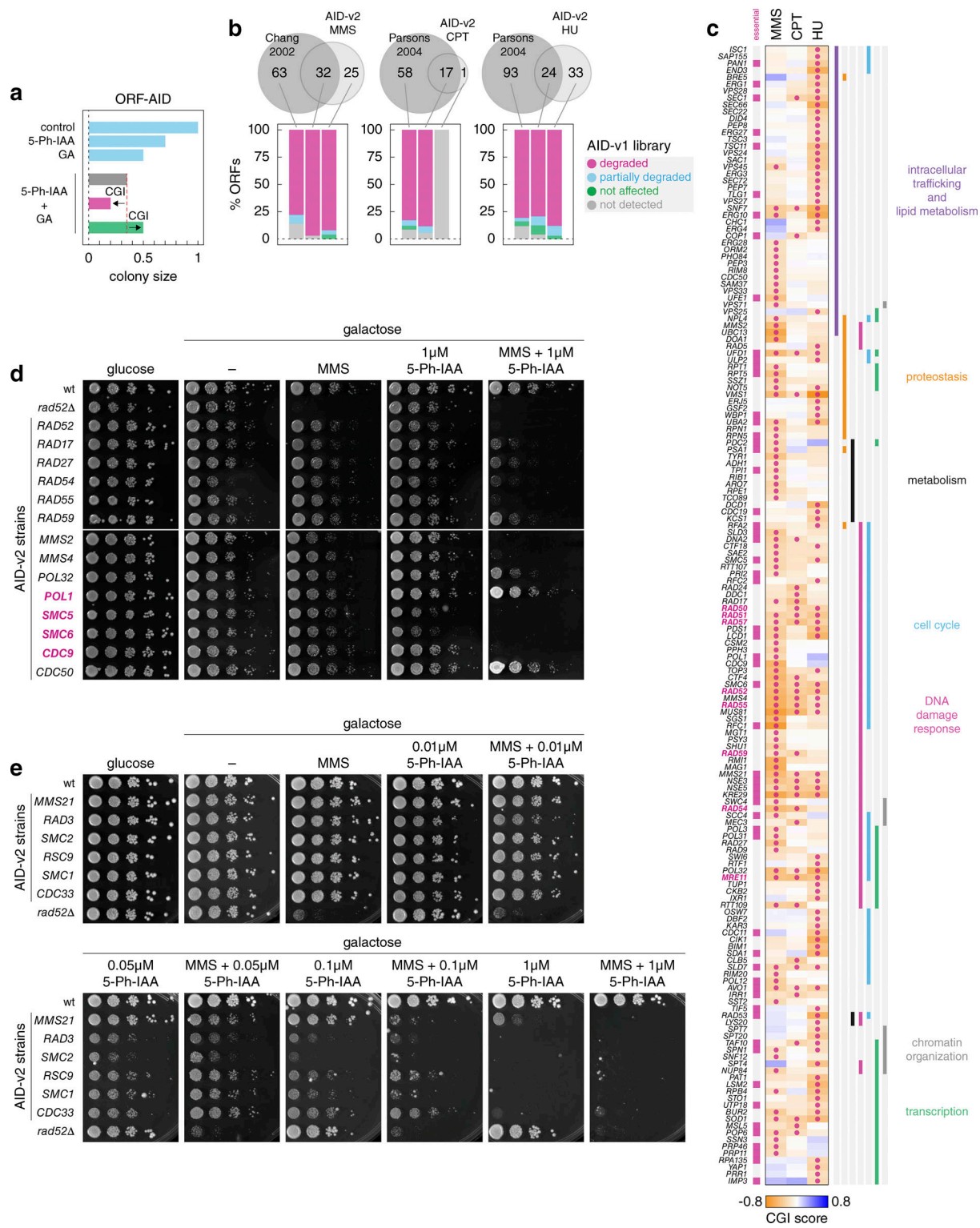

Figure S5. **Genome-wide screens for DNA damage response factors. (a)** Definition of CGI. Fitness of an ORF-AID strain (normalized colony size, Materials and methods) measured under four conditions: control (no perturbation), in the presence of 5-Ph-IAA (to induce degradation of the AID-tagged protein), GA, or both. Grey bar, expected fitness in the double perturbation condition (product of the fitness from the single perturbation conditions) in the absence of a CGI. Magenta and green bars, examples of negative and positive CGIs. **(b)** Top, overlap between chemical-genetic interactors identified with the AID-v2 library and previously identified resistance genes for MMS (Chang et al., 2002), CPT, or HU (Parsons et al., 2004). Essential factors identified with the AID-v2 library were excluded from the comparison. Bottom, frequency of *OsTIR1*-dependent degradation phenotypes in the AID-v1 library for the indicated sets of genes. **(c)** Heatmap of negative chemical-genetic interactors with MMS, CPT, or HU, as in Fig. 4 d. Gene names, essential genes and significant interactors (circles) with each GA are indicated. **(d and e)** 10-fold serial dilutions of the indicated strains from the AID-v2 library, a wild type (wt) and a *RAD52* knockout on the indicated media. Essential genes are marked in magenta (d).

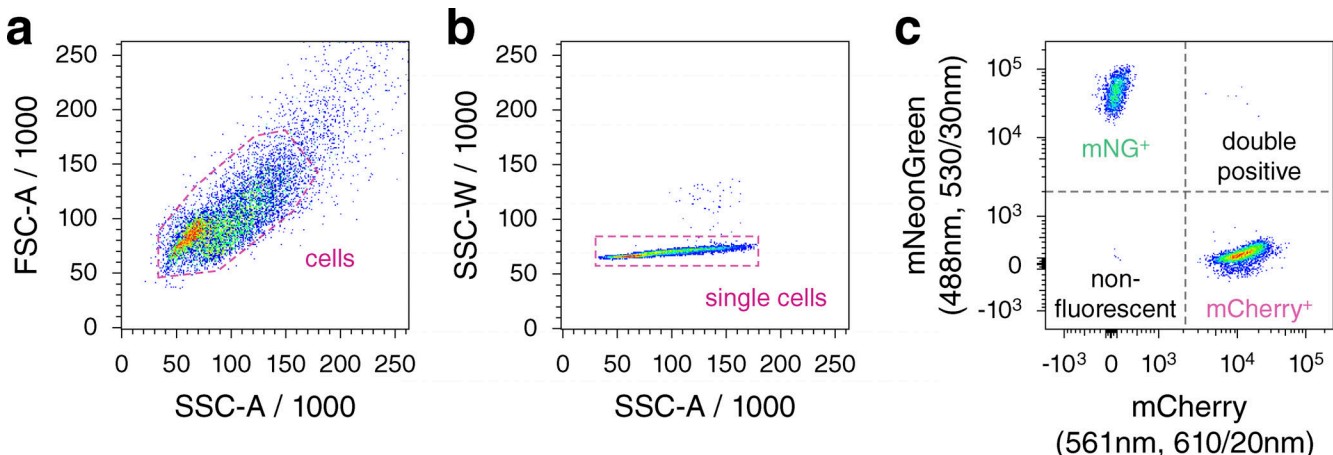

Figure S6. **Representative gating strategy used in flow cytometry experiments. (a)** Measurements were gated to identify cells based on size and granularity using forward and side scatter areas. **(b)** Cells were subsequently gated for single cells using side scatter area and width. **(c)** Finally, single cells were gated into four populations based on their mNG or mCherry fluorescence intensities compared to the maximum intensities of a non-fluorescent control population.

**Provided online are Table S1, Table S2, Table S3, Table S4, Table S5, Table S6, and Table S7. Table S1 shows results of the genome-wide competition screen. Table S2 shows terminal localization signals in the yeast proteome. Table S3 shows characterization of the AID-v1 library. Table S4 shows results of the genome-wide screens for DNA damage response factors. Table S5 shows gene set enrichment analysis of resistance factors unique to each GA. Table S6 shows yeast strains used in this study. Table S7 shows plasmids used in this study.**

