## [Peer Review File · The Journal of Cell Biology]

Genome-wide conditional degron libraries for functional genomics

Eduardo Gameiro, Karla Juárez-Núñez, Jia Jun Fung, Susmitha Shankar, Brian Luke, and Anton Khmelinskii

Corresponding Author(s): Anton Khmelinskii, Institute of Molecular Biology

Review Timeline:

Submission Date:	2024-09-03
Editorial Decision:	2024-10-30
Revision Received:	2024-11-06

Monitoring Editor: Jodi Nunnari

Scientific Editor: Dan Simon

Transaction Report:

DOI: <https://doi.org/10.1083/jcb.202409007>

Revision 0

Review #1

1. Evidence, reproducibility and clarity:

Evidence, reproducibility and clarity (Required)

****Summary****

The manuscript by the Khmelinskii group reports that they have successfully constructed two conditional degron libraries of budding yeast for almost all proteins. For this purpose, the authors employed an improved auxin-inducible degron (AID2). Initially, they constructed yeast libraries by fusing HaloTag to the N- or C-terminus of proteins and found that C-terminal tagging is less likely to affect the location and function of proteins (Fig. 1). Based on this finding, the authors fused mNG-AID*-3Myc or AID*-3Myc (AID-v1 or AID-v2 library, respectively) to more than 5600 proteins and found that 4079 proteins were significantly depleted when cells were treated with 5-Ph-IAA (Fig. 2). A fitness defect was observed for over 60% of essential proteins, indicating the target depletion showed the expected phenotype in many cases (Fig. 3). Finally, the authors screened proteins required for maintaining viability in the presence of MMS, CPT and HU, and identified common proteins involved in DNA repair (such as RAD52 epistasis proteins) and other proteins specific for MMS, CPT or HU resistance (Fig. 4). Furthermore, the authors revealed that an ER membrane protein, Gsf2, is required for HU resistance, which was not found in previous studies with the YKO library because *gsf2Δ* cells in the YKP library had acquired a suppressor mutation (Fig. 4e).

****Major comments****

- In Figure S2a, the authors initially checked the growth of yeast cells expressing OsTIR1(F74G) under the GAL1 promoter, saying that "expression of OsTir1(F74G) from the strong galactose-inducible GAL1 promoter had a negligible impact on yeast fitness (page 3)". To me, the OsTIR1(G74G) expressing cells showed slightly slower growth compared to the control cells. Moreover, the cells expressing it under the very strong GPD promoter showed apparent slow growth, suggesting that OstIR1(F74G) overexpression caused a side effect. The authors should carefully evaluate the cells with GAL1-OsTIR1(F74G).
- Given the possibility that OsTIR1(F74G) overexpression might cause a growth problem, it is not appropriate to compare OsTIR1+ and OsTIR1- conditions for evaluating growth fitness (Fig. 2). As shown in Fig. S4b, it is more appropriate to compare the +/- 5-Ph-IAA conditions. Additionally, the 5-Ph-IAA concentration used in this study was not clearly mentioned in the method section and figure legends.
- The authors found that fitness defects were observed for over 60% of essential proteins (Fig. 3). In other words, depletion of the remaining 40% was not enough to induce growth defects. The authors should discuss how the current AID library can be improved to achieve better target depletion. Previous literature reported various possibilities, such as using a tandem degron tag and combining AID with the Tet promoter system (PMID 25181302, 26081484). Although optional, it would be wonderful if the authors would generate an improved library.

****Minor comments****

- 5-Ph-IAA is not auxin because it does not induce the auxin responses in plants (PMID 29355850). Therefore, the authors should be careful when they refer to 5-Ph-IAA and should not call it auxin.
- The availability of the HaloTag and AID libraries should be indicated.
- Page 3: "Finally, the extent of AID-dependent degradation varied with protein abundance, in that highly expressed proteins were more likely to be only partially degraded compared to lowly expressed ones (Fig. 2e, Fig. S2e)". Fig. S2e should be Fig. S2d, shouldn't it?

2. Significance:

Significance (Required)

This paper is technically robust and well-conducted. It presents a comprehensive study showcasing the effectiveness of the conditional degron library. The HaloTag libraries will also be useful. The yeast libraries presented in this study will be invaluable for future screenings and studies across all aspects of yeast biology.

3. How much time do you estimate the authors will need to complete the suggested revisions:

Estimated time to Complete Revisions (Required)

(Decision Recommendation)

Between 1 and 3 months

4. Review Commons values the work of reviewers and encourages them to get credit for their work. Select 'Yes' below to register your reviewing activity at Web of Science Reviewer Recognition Service (formerly Publons); note that the content of your review will not be visible on Web of Science.

Yes

Review #2

1. Evidence, reproducibility and clarity:

Evidence, reproducibility and clarity (Required)

In this study, the authors reported the development and characterization of two AID-tagged strain libraries for the model organism *S. cerevisiae*. The libraries are based on the latest AID technology, AID2. One library contains a fluorescent protein fused to the AID, whereas the other library does not have the fluorescent protein, thus offering better compatibility with imaging-based screens. The authors show that AID-dependent protein degradation can be achieved for most of the library strains, and a growth phenotype was induced for a high fraction of essential genes. Genetic screens for DNA damage-sensitive mutants showcased the applicability of the libraries.

I only have the following minor comments and suggestions for the authors to consider.

Point 1

Page 3

"Optimized tagging of proteins with these N-terminal localization signals likely also contributes to the lack of correlation between differential fitness defects and occurrence of terminal localization signals (Fig. S1f, Table S2). "

Is this because the genes that cannot tolerate C-terminal addition are already depleted in the C-SWAT library? In the C-SWAT library, a 15-amino-acid linker L3 is added to the C-terminus.

Point 2

Page 3

"Expression of OsTir1(F74G) from the strong galactose-inducible GAL1 promoter had a negligible impact on yeast fitness (Fig. S2a)."

I wonder why the authors chose to use an inducible promoter to express OsTir1(F74G). In other studies, for example Snyder et al. 2019, OsTir1 has been expressed from a constitutive promoter.

Point 3

Page 3

"A similar frequency was previously observed with a set of AID alleles constructed for 758 essential ORFs using the original AID system (Snyder et al, 2019). However, over a third of these alleles exhibited fitness defects even in the absence of auxin, which were further compounded by off-target effects of auxin, highlighting the advantages of the AID2 system."

Snyder et al. 2019 used a TAP-AID-6FLAG tag. The fitness defect in the absence of auxin may not necessarily be due to the AID part of the tag, as TAP tagging is known to compromise the functions of some genes.

Point 4

Page 3

"Interestingly, complete degradation of 33% of essential proteins did not result in a fitness defect. It is possible that in some cases partial degradation results in low protein levels that are below the detection limit of our assay but are sufficient for viability."

Are these "33% of essential proteins" enriched with genes with low expression levels? I guess genes with low expression levels are more likely to fall below the detection limit even when partially depleted. Are there extreme examples where a highly expressed essential gene does not exhibit a fitness defect when the protein product is no longer detectable?

2. Significance:

Significance (Required)

This study generated highly valuable resources for functional genomic studies.

3. How much time do you estimate the authors will need to complete the suggested revisions:

Estimated time to Complete Revisions (Required)

(Decision Recommendation)

Less than 1 month

4. Review Commons values the work of reviewers and encourages them to get credit for their work. Select 'Yes' below to register your reviewing activity at Web of Science Reviewer Recognition Service (formerly Publons); note that the content of your review will not be visible on Web of Science.

Yes

Review #3

1. Evidence, reproducibility and clarity:

Evidence, reproducibility and clarity (Required)

Summary: In this manuscript, the authors construct and analyze a genome-wide collection of AID-tagged *S. cerevisiae* strains. The manuscript is clearly written and the analysis appears to be thorough. This collection will be quite useful to the yeast community. There are some issues to address, listed below.

1. page 1, abstract - "...with protein abundance and tag accessibility as limiting factors." It's not clear what the authors mean by protein abundance as a limiting factor. Are they referring to the protein level pre-depletion? Please clarify.
2. page 1, second paragraph of the Intro, end of the paragraph - There are publications prior to Van Leeuwen et al. 2016 that describe suppressors lurking in the deletion set. Here are two that should be cited: Hughes et al. 2000, DOI: 10.1038/77116 and Teng et al. 2013, DOI: 10.1038/77116 .
3. The goal of this work, stated in the first sentence of results, is to construct genome-wide AID libraries. Yet, to test whether N-terminal or C-terminal tagging is better, the authors used a Halo tag. Those results showed that, for the Halo tag, C-terminal tagging was less likely to impair function. Why weren't these tests done with the same AID tag used to build the libraries in the next section? What is the evidence that the results for a Halo tag will be the same as for an AID tag? While hard to find it documented in publications, there is a lot of anecdotal evidence that the type of tag can make a big difference, as well as its location. While this section will be of interest to those using Halo tags, it's not clear how it relates to the rest of the paper, especially given the careful characterization of the AID library in the next section.

4. Throughout this manuscript, including the Tables, in cases where the protein is no longer detected, please do not describe this as "complete degradation." Instead, please use "not detectable." This is clearly the case for essential proteins that are no longer detected but that still grow, so it is very likely the case for many or all of the others. If the authors have any understanding of the sensitivity of their fluorescence assay, then that would be helpful to know. For example, they could add a control, taking a known amount of a fluorescent protein and analyzing known dilutions to assay the level of detection.

5. Fig. S2a and page 3, first full paragraph - The authors wrote that expression of OsTir1(F74G) from the strong GAL1 promoter had a negligible impact on growth. However, the figure shows that there is an obvious effect on growth after 48 hours of incubation, with much smaller colonies. This defect is much less obvious after 72 hours. This difference suggests that the growth effect would have been even more obvious at 24 hours. I think that the text should be modified to indicate this effect.

6. One of the main justifications for the construction of the AID library is to allow assays for essential genes. Yet that was not a feature of the screen for DNA damage response factors. Were any essential genes identified in those screens? It would be of great interest to identify lower levels of 5-Ph-IAA that only mildly affect growth of essential genes and then to repeat the screens.

7. A big advantage of AID depletion over deletions is the ability to look at strains very shortly after loss of the protein of interest. In many cases in the literature, experiments are done after one or two hours after depletion. Yet in this work, there are no data presented on how effective depletion is in the short term versus after a long period of growth (24 hours). It would be a strong addition to the manuscript to include a time course for at least a subset of the proteins to look at the loss of signal over time, either by fluorescence or by Western blots.

2. Significance:

Significance (Required)

The library will be of use to the yeast community.

3. How much time do you estimate the authors will need to complete the suggested revisions:

Estimated time to Complete Revisions (Required)

(Decision Recommendation)

Between 1 and 3 months

4. Review Commons values the work of reviewers and encourages them to get credit for their work. Select 'Yes' below to register your reviewing activity at Web of Science Reviewer Recognition Service (formerly Publons); note that the content of your review will not be visible on Web of Science.

Yes

Manuscript number: RC-2024-02550

Corresponding author(s): Anton Khmelinskii

1. General Statements

We thank all reviewers for their constructive criticism and suggestions. We have addressed all the points as detailed below. We also added an experiment that strengthens the connection between replication stress and *GSF2* and suggests a role of *GSF2* in recovery from the DNA replication checkpoint arrest (Fig. 4g).

Reviewer #1 (Evidence, reproducibility and clarity)

Summary

The manuscript by the Khmelinskii group reports that they have successfully constructed two conditional degron libraries of budding yeast for almost all proteins. For this purpose, the authors employed an improved auxin-inducible degron (AID2). Initially, they constructed yeast libraries by fusing HaloTag to the N- or C-terminus of proteins and found that C-terminal tagging is less likely to affect the location and function of proteins (Fig. 1). Based on this finding, the authors fused mNG-AID*-3Myc or AID*-3Myc (AID-v1 or AID-v2 library, respectively) to more than 5600 proteins and found that 4079 proteins were significantly depleted when cells were treated with 5-Ph-IAA (Fig. 2). A fitness defect was observed for over 60% of essential proteins, indicating the target depletion showed the expected phenotype in many cases (Fig. 3). Finally, the authors screened proteins required for maintaining viability in the presence of MMS, CPT and HU, and identified common proteins involved in DNA repair (such as RAD52 epistasis proteins) and other proteins specific for MMS, CPT or HU resistance (Fig. 4). Furthermore, the authors revealed that an ER membrane protein, Gsf2, is required for HU resistance, which was not found in previous studies with the YKO library because *gsf2* Δ cells in the YKP library had acquired a suppressor mutation (Fig. 4e).

Major comments

1 - In Figure S2a, the authors initially checked the growth of yeast cells expressing OsTIR1(F74G) under the GAL1 promoter, saying that "expression of OsTir1(F74G) from the strong galactose-inducible GAL1 promoter had a negligible impact on yeast fitness (page 3)". To me, the OsTIR1(G74G) expressing cells showed slightly slower growth compared to the control cells. Moreover, the cells expressing it under the very strong GPD promoter showed apparent slow growth, suggesting that OsTIR1(F74G) overexpression caused a side effect. The authors should carefully evaluate the cells with GAL1-OsTIR1(F74G).

Indeed high levels of OsTir1(F74G) impaired growth, at least in the strain background used in our experiments. Expression from the strongest promoter we tested (*GPD*) resulted in an obvious fitness defect, whereas conditional expression from the strong *GAL1* promoter had a small impact on fitness and expression from the weaker *CYC1* and *ADH1* promoters did not affect fitness (Fig. S2a). Despite the small fitness impairment, we decided to use the *GAL1-OsTIR1(F74G)* construct for the AID libraries for two reasons: the conditional nature of this promoter is likely to limit adaptation to expression of OsTir1(F74G) and the high expression levels of OsTir1(F74G) are less likely to limit degradation of AID-tagged proteins. We added this explanation to the Results section.

As suggested by the reviewer, we quantitatively evaluated the fitness impact of the *GAL1-OsTIR1(F74G)* construct. Using the colony size data of the AID-v1 library (grown on galactose medium with 1 μ M 5-Ph-IAA, Fig. 2c), we compared colony sizes of *OsTIR1⁻* and *OsTIR1⁺* strains for non-essential ORFs. As degradation of non-essential proteins is not expected to affect fitness, the difference in colony size between *OsTIR1⁻* and *OsTIR1⁺* strains can be attributed to OsTir1 expression. On average, the presence of the *OsTIR1* construct reduced colony size by 7% (median fitness of *OsTIR1⁺* strains relative to *OsTIR1⁻* strains of 0.93 ± 0.06 , $n = 4698$ non-essential ORFs). We performed the same comparison for strains that did not exhibit OsTIR1-dependent protein degradation. In this set of strains, the presence of the *OsTIR1* construct also reduced colony size by 7% (median fitness of *OsTIR1⁺* strains relative to *OsTIR1⁻* strains of 0.93 ± 0.05 , $n = 624$ ORFs in the “not affected” group in Fig. 2d). We added this information to Fig. S3a.

2 - Given the possibility that OsTIR1(F74G) overexpression might cause a growth problem, it is not appropriate to compare *OsTIR1⁺* and *OsTIR1⁻* conditions for evaluating growth fitness (Fig. 2). As shown in Fig. S4b, it is more appropriate to compare the +/- 5-Ph-IAA conditions. Additionally, the 5-Ph-IAA concentration used in this study was not clearly mentioned in the method section and figure legends.

The two approaches, comparison of *OsTIR1⁻* and *OsTIR1⁺* strains grown on galactose with 5-Ph-IAA (as was done for the AID-v1 library) and comparison of galactose \pm 5-Ph-IAA conditions (as was done for the AID-v2 library), have advantages and disadvantages but should yield similar results. The technical noise (due to spatial effects on the screen plates) is lower for the comparison of *OsTIR1⁻* and *OsTIR1⁺* strains, as the two strains for each ORF can be grown next to each other on the same plate (Fig. 2c). Furthermore, corrections of spatial effects are more precise with this layout as the frequency of fitness defects per plate is lower. On the other hand, comparison of galactose \pm 5-Ph-IAA conditions implicitly corrects for the fitness impact of the *GAL1-OsTIR1(F74G)* construct, as the fitness distribution of each condition is normalized to the median of that condition, but this fitness impact of OsTir1 cannot be determined from the screen results.

We now explicitly corrected the colony size data of the AID-v1 library for the fitness impact of OsTir1 expression (quantified in the previous point) and updated all the analyses and results shown in Fig. 3, Fig. S3b-e and Fig. S4a. The correction was performed using the multiplicative

model, whereby the fitness impacts of OsTir1 expression and degradation of the AID-tagged protein are independent. Overall, our observations and conclusions stand unchanged with the corrected data.

Finally, the 5-Ph-IAA concentration (1 μ M) used in all experiments is now indicated in the figure legends and the Methods section.

3 - The authors found that fitness defects were observed for over 60% of essential proteins (Fig. 3). In other words, depletion of the remaining 40% was not enough to induce growth defects. The authors should discuss how the current AID library can be improved to achieve better target depletion. Previous literature reported various possibilities, such as using a tandem degron tag and combining AID with the Tet promoter system (PMID 25181302, 26081484). Although optional, it would be wonderful if the authors would generate an improved library.

Following the reviewer's suggestion, we added the following statement to the discussion:

"In the future, the libraries could be potentially improved with N-terminal tagging of ORFs that currently exhibit incomplete or no degradation of AID-tagged proteins or using multiple copies of the AID* tag to enhance protein degradation (Kubota et al, 2013; Nishimura & Kanemaki, 2014)."

Minor comments

4 - 5-Ph-IAA is not auxin because it does not induce the auxin responses in plants (PMID 29355850). Therefore, the authors should be careful when they refer to 5-Ph-IAA and should not call it auxin.

We corrected this and now refer to 5-Ph-IAA explicitly throughout the manuscript.

5 - The availability of the HaloTag and AID libraries should be indicated.

We added the following statement to the Methods section: "All strains, plasmids and libraries are available upon request."

6 - Page 3: "Finally, the extent of AID-dependent degradation varied with protein abundance, in that highly expressed proteins were more likely to be only partially degraded compared to lowly expressed ones (Fig. 2e, Fig. S2e)". Fig. S2e should be Fig. S2d, shouldn't it?

We corrected this mistake.

Reviewer #1 (Significance):

This paper is technically robust and well-conducted. It presents a comprehensive study showcasing the effectiveness of the conditional degron library. The HaloTag libraries will also be useful. The yeast libraries presented in this study will be invaluable for future screenings and studies across all aspects of yeast biology.

Reviewer #2 (Evidence, reproducibility and clarity):

In this study, the authors reported the development and characterization of two AID-tagged strain libraries for the model organism *S. cerevisiae*. The libraries are based on the latest AID technology, AID2. One library contains a fluorescent protein fused to the AID, whereas the other library does not have the fluorescent protein, thus offering better compatibility with imaging-based screens. The authors show that AID-dependent protein degradation can be achieved for most of the library strains, and a growth phenotype was induced for a high fraction of essential genes. Genetic screens for DNA damage-sensitive mutants showcased the applicability of the libraries.

I only have the following minor comments and suggestions for the authors to consider.

Point 1, Page 3

"Optimized tagging of proteins with these N-terminal localization signals likely also contributes to the lack of correlation between differential fitness defects and occurrence of terminal localization signals (Fig. S1f, Table S2)."

Is this because the genes that cannot tolerate C-terminal addition are already depleted in the C-SWAT library? In the C-SWAT library, a 15-amino-acid linker L3 is added to the C-terminus.

That is certainly a possibility. During construction of SWAT library, tagging with N-SWAT and C-SWAT acceptor modules failed for 251 and 353 ORFs, respectively (Weill et al. 2018, Meurer et al. 2018). However, these ORFs are not enriched in N- or C-terminal localization signals, respectively (4.6% ORFs with C-terminal signals in C-SWAT library vs 3.3% among failed C-SWAT strains; 12.3% ORFs with N-terminal signals in N-SWAT library vs 2.0% among failed N-SWAT strains).

The most significant trend in the data is enrichment of ribosomal subunits in both sets of failed strains: 3.9% and 16.3% of the genes mapped to the GO term "ribosome" in the N-SWAT library and the set of failed N-SWAT strains, respectively; 3.6% and 15.9% of the genes in the C-SWAT library and the set of failed C-SWAT strains, respectively. This is consistent with what was reported by Weill et al. for failed N-SWAT strains.

Point 2, Page 3

"Expression of *OsTir1(F74G)* from the strong galactose-inducible *GAL1* promoter had a negligible impact on yeast fitness (Fig. S2a)."

I wonder why the authors chose to use an inducible promoter to express *OsTir1(F74G)*. In other studies, for example Snyder et al. 2019, *OsTir1* has been expressed from a constitutive promoter.

Despite the small fitness impairment, we decided to use the *GAL1-OsTIR1(F74G)* construct for the AID libraries for two reasons: the conditional nature of this promoter is likely to limit adaptation to expression of *OsTir1(F74G)* and the high expression levels of *OsTir1(F74G)* are less likely to limit degradation of AID-tagged proteins. We added this explanation to the Results section.

Please see our response to reviewer 1, points 1 and 2.

Point 3, Page 3

"A similar frequency was previously observed with a set of AID alleles constructed for 758 essential ORFs using the original AID system (Snyder et al, 2019). However, over a third of these alleles exhibited fitness defects even in the absence of auxin, which were further compounded by off-target effects of auxin, highlighting the advantages of the AID2 system."

Snyder et al. 2019 used a TAP-AID-6FLAG tag. The fitness defect in the absence of auxin may not necessarily be due to the AID part of the tag, as TAP tagging is known to compromise the functions of some genes.

We corrected our statement as follows:

"A similar frequency was previously observed with a set of AID alleles constructed for 758 essential ORFs using the original AID system (Snyder et al, 2019). However, over a third of these alleles exhibited fitness defects even in the absence of auxin, which were further compounded by off-target effects of auxin."

Point 4, Page 3

"Interestingly, complete degradation of 33% of essential proteins did not result in a fitness defect. It is possible that in some cases partial degradation results in low protein levels that are below the detection limit of our assay but are sufficient for viability."

Are these "33% of essential proteins" enriched with genes with low expression levels? I guess genes with low expression levels are more likely to fall below the detection limit even when partially depleted. Are there extreme examples where a highly expressed essential gene does not exhibit a fitness defect when the protein product is no longer detectable?

We performed the analysis suggest by the reviewer, and observed no difference in pre-degradation protein levels between essential & degraded proteins with and without a fitness defect (now shown in Fig. S3b). This also showed that there are indeed several essential proteins with high pre-degradation proteins levels and without a fitness defects upon degradation to below our detection limit: Pgi1, Nhp2, Smt3, Gus1, Dys1, Sis1, Fas2 and Rpo26 (in the abundance bin 4 in Fig. S2f).

In addition, we considered the nature of the essential genes in these two groups. Namely, we compared the frequency of core essential genes, which are always required for viability, and conditional essential genes, which vary in essentiality depending on the genetic background or environment (Bosch-Guiteras & van Leeuwen, 2022). Interestingly, the set of essential and degraded proteins without an accompanying fitness defect was enriched in conditional essential genes defined by two independent measures: essentiality across *S. cerevisiae* natural isolates (Peter et al, 2018) or with bypass suppression interactions in a laboratory strain (van Leeuwen et al, 2020) (Fig. S3c, odds ratio = 1.6, p-value = 0.04 in a Fisher's exact test and odds ratio = 1.7, p-value = 0.02, respectively). This suggests that conditional essentiality could explain the observed lack of fitness defects upon degradation of some essential proteins.

Full Revision

We added this analysis to the Results section.

Reviewer #2 (Significance):

This study generated highly valuable resources for functional genomic studies.

Reviewer #3 (Evidence, reproducibility and clarity):

Summary: In this manuscript, the authors construct and analyze a genome-wide collection of AID-tagged *S. cerevisiae* strains. The manuscript is clearly written and the analysis appears to be thorough. This collection will be quite useful to the yeast community. There are some issues to address, listed below.

1. page 1, abstract - "...with protein abundance and tag accessibility as limiting factors." It's not clear what the authors mean by protein abundance as a limiting factor. Are they referring to the protein level pre-depletion? Please clarify.

That is correct. We clarified this statement as follows:

"Almost 90% of AID-tagged proteins were degraded in the presence of the auxin analog 5-Ph-IAA, with initial protein abundance and tag accessibility as limiting factors."

2. page 1, second paragraph of the Intro, end of the paragraph - There are publications prior to Van Leeuwen et al. 2016 that describe suppressors lurking in the deletion set. Here are two that should be cited: Hughes et al. 2000, DOI: 10.1038/77116 and Teng et al. 2013, DOI: 10.1016/j.molcel.2013.09.026.

We added the references pointed out by the review.

3. The goal of this work, stated in the first sentence of results, is to construct genome-wide AID libraries. Yet, to test whether N-terminal or C-terminal tagging is better, the authors used a Halo tag. Those results showed that, for the Halo tag, C-terminal tagging was less likely to impair function. Why weren't these tests done with the same AID tag used to build the libraries in the next section? What is the evidence that the results for a Halo tag will be the same as for an AID tag? While hard to find it documented in publications, there is a lot of anecdotal evidence that the type of tag can make a big difference, as well as its location. While this section will be of interest to those using Halo tags, it's not clear how it relates to the rest of the paper, especially given the careful characterization of the AID library in the next section.

We chose the Halo tag due its size (33 kDa), similar to many commonly used fluorescent protein tags and to the mNG-AID*-3myc tag in the AID-v1 library, and lack of evidence for a dominant negative effect on the tagged proteins. This is now stated in the Results section.

We agree that further work is needed to understand how the type of tag, its size and biophysical properties, and the linker between the tag and the protein of interest affect protein localization and function across the proteome. This is now stated in the Results section.

4. Throughout this manuscript, including the Tables, in cases where the protein is no longer detected, please do not describe this as "complete degradation." Instead, please use "not detectable." This is clearly the case for essential proteins that are no longer detected but that still grow, so it is very likely the case for many or all of the others. If the authors have any understanding of the sensitivity of their fluorescence assay, then that would be helpful to know. For example, they could add a control, taking a known amount of a fluorescent protein and analyzing known dilutions to assay the level of detection.

We appreciate the reviewer's suggestion. We decided against "not detectable" instead of "complete degradation" to avoid confusion with proteins that are not detectable pre-degradation. Nevertheless, we replaced "complete degradation" with "degradation" and added the following explanation to the Results section:

"Out of 5079 proteins detected in *OsTIR1*⁻ strains, 4455 (~88%) were significantly depleted in *OsTIR1*⁺ strains (Fig. 2d, Table S3). 3981 proteins could not be detected specifically in the *OsTIR1*⁺ background. Hereafter, we will refer to these proteins as degraded, although it is likely that at least in some cases degradation is not complete but the remainder is below the detection limit of our plate reader assay. Nevertheless, 474 proteins were unequivocally degraded only partially, as they were detectable in the *OsTIR1*⁺ background but at reduced levels compared to the *OsTIR1*⁻ background (Fig. 2d)."

To estimate the detection limit of the colony fluorescence assay, we correlated the background-corrected mNG intensities in *OsTIR1*⁻ strains with absolute levels (in molecules per cell) of 1167 proteins determined by Lawless et al. (PMID 26750110). Based on a linear fit, the threshold above which proteins are considered "detected" in our analysis, $mNG/bkg(OsTIR1^-) > 1.2$, corresponds to 200 molecules per cell (95% confidence interval 18 to 2187 molecules per cell). We added this information to the Results section and Fig. S2c.

This detection limit is in line with our results, where low abundance proteins such as the centromeric histone Cse4/CENP-A (with two Cse4 molecules per centromere adding to 64 molecules per cell, Aravamudhan et al. PMID: 23623551 and several times that amount elsewhere in the cell, Collins et al. PMID: 15530401) can be detected in the colony assay (Table S3).

5. Fig. S2a and page 3, first full paragraph - The authors wrote that expression of *OsTir1*(F74G) from the strong GAL1 promoter had a negligible impact on growth. However, the figure shows that there is an obvious effect on growth after 48 hours of incubation, with much smaller colonies. This defect is much less obvious after 72 hours. This difference suggests that the growth effect would have been even more obvious at 24 hours. I think that the text should be modified to indicate this effect.

We now quantified the fitness impact of the *GAL1-OsTIR1(F74G)* construct and rephrased this part of the manuscript. In addition, we corrected the AID-v1 library screen results for the fitness impact of the *GAL1-OsTIR1(F74G)* construct and updated all figures and tables. Please see our response to reviewer 1, points 1 and 2.

6. One of the main justifications for the construction of the AID library is to allow assays for essential genes. Yet that was not a feature of the screen for DNA damage response factors. Were any essential genes identified in those screens? It would be of great interest to identify lower levels of 5-Ph-IAA that only mildly affect growth of essential genes and then to repeat the screens.

58 out of the combined 165 potential resistance factors identified in the three screens are essential genes. We added this information to the Results section and essential genes are now indicated in Fig. S5c.

We now show that chemical-genetic interactions for both essential and non-essential genes can be reproduced in spot tests using the MMS screen as an example (Fig. S5d). We also show that additional essential hits can be identified at lower concentrations of 5-Ph-IAA, which allow determining chemical-genetic interactions for strains that otherwise exhibit no growth in 1 μ M 5-Ph-IAA (Fig. S5e). As the screens serve as a demonstration of possible uses of the AID libraries, we consider additional exhaustive screening for DNA damage response factors beyond the scope of this manuscript.

7. A big advantage of AID depletion over deletions is the ability to look at strains very shortly after loss of the protein of interest. In many cases in the literature, experiments are done after one or two hours after depletion. Yet in this work, there are no data presented on how effective depletion is in the short term versus after a long period of growth (24 hours). It would be a strong addition to the manuscript to include a time course for at least a subset of the proteins to look at the loss of signal over time, either by fluorescence or by Western blots.

We performed time courses of protein depletion with immunoblotting for 12 strains (4 proteins from the “degraded”, “partially degraded” and “not affected” groups each). The results in Fig. S2e show that “degraded” proteins are depleted to below the detection limit within 60min of 5-Ph-IAA addition, “partially degraded” proteins are depleted less or exhibit a degradation-resistant pool, and the levels of “not affected” proteins remain stable over time, consistent with their classification based on mNG fluorescence in the colony assay. We added this information to the Results section.

Reviewer #3 (Significance):

The library will be of use to the yeast community.

October 30, 2024

RE: JCB Manuscript #202409007T

Dr. Anton Khmelinskii
Institute of Molecular Biology
Ackermannweg 4
Mainz 55128
Germany

Dear Dr. Khmelinskii,

Thank you for submitting your revised manuscript entitled "Genome-wide conditional degron libraries for functional genomics." We would be happy to publish your paper in JCB pending final revisions necessary to meet our formatting guidelines (see details below).

A. MANUSCRIPT ORGANIZATION AND FORMATTING:

1) Text limits: Character count for Tools is < 40,000, not including spaces. Count includes title page, abstract, introduction, results, discussion, and acknowledgments. Count does not include materials and methods, figure legends, references, tables, or supplemental legends.

2) Figure formatting: Tools may have up to 10 main text figures. Scale bars must be present on all microscopy images, including inset magnifications. Molecular weight or nucleic acid size markers must be included on all gel electrophoresis. Please add MW markers to blots in Figure S2E.

Also, please avoid pairing red and green for images and graphs to ensure legibility for color-blind readers. If red and green are paired for images, please ensure that the particular red and green hues used in micrographs are distinctive with any of the colorblind types. If not, please modify colors accordingly or provide separate images of the individual channels.

3) Statistical analysis: Error bars on graphic representations of numerical data must be clearly described in the figure legend. The number of independent data points (n) represented in a graph must be indicated in the legend. Please, indicate whether 'n' refers to technical or biological replicates (i.e. number of analyzed cells, samples or animals, number of independent experiments). If independent experiments with multiple biological replicates have been performed, we recommend using distribution-reproducibility SuperPlots (please see Lord et al., JCB 2020) to better display the distribution of the entire dataset, and report statistics (such as means, error bars, and P values) that address the reproducibility of the findings.

Statistical methods should be explained in full in the materials and methods. For figures presenting pooled data the statistical measure should be defined in the figure legends. Please also be sure to indicate the statistical tests used in each of your experiments (both in the figure legend itself and in a separate methods section) as well as the parameters of the test (for example, if you ran a t-test, please indicate if it was one- or two-sided, etc.). Also, if you used parametric tests, please indicate if the data distribution was tested for normality (and if so, how). If not, you must state something to the effect that "Data distribution was assumed to be normal but this was not formally tested."

4) Materials and methods: Should be comprehensive and not simply reference a previous publication for details on how an experiment was performed. Please provide full descriptions (at least in brief) in the text for readers who may not have access to referenced manuscripts. The text should not refer to methods "...as previously described." Please also indicate the type of membrane used for immunoblotting as well as describe acquisition and quantification methods.

5) For all cell lines, vectors, constructs/cDNAs, etc. - all genetic material: please include database / vendor ID (e.g., Addgene, ATCC, etc.) or if unavailable, please briefly describe their basic genetic features, even if described in other published work or gifted to you by other investigators (and provide references where appropriate). Please be sure to provide the sequences for all of your oligos: primers, si/shRNA, RNAi, gRNAs, etc. in the materials and methods. You must also indicate in the methods the source, species, and catalog numbers/vendor identifiers (where appropriate) for all of your antibodies, including secondary. If antibodies are not commercial, please add a reference citation if possible.

- 6) Microscope image acquisition: The following information must be provided about the acquisition and processing of images:
- Make and model of microscope
 - Type, magnification, and numerical aperture of the objective lenses
 - Temperature
 - Imaging medium
 - Fluorochromes
 - Camera make and model
 - Acquisition software
 - Any software used for image processing subsequent to data acquisition. Please include details and types of operations involved (e.g., type of deconvolution, 3D reconstitutions, surface or volume rendering, gamma adjustments, etc.).
- 7) References: There is no limit to the number of references cited in a manuscript. References should be cited parenthetically in the text by author and year of publication. Abbreviate the names of journals according to PubMed.
- 8) Supplemental materials: Tools generally may have up to 5 supplemental figures and 10 videos. You currently exceed this limit but, in this case, we will be able to give you the extra space. Please also note that tables, like figures, should be provided as individual, editable files. A summary of all supplemental material should appear at the end of the Materials and methods section. Please include one brief sentence per item.
- 9) eTOC summary: A ~40-50 word summary that describes the context and significance of the findings for a general readership should be included on the title page. The statement should be written in the present tense and refer to the work in the third person. It should begin with "First author name(s) et al..." to match our preferred style.
- 10) Conflict of interest statement: JCB requires inclusion of a statement in the acknowledgements regarding competing financial interests. If no competing financial interests exist, please include the following statement: "The authors declare no competing financial interests." If competing interests are declared, please follow your statement of these competing interests with the following statement: "The authors declare no further competing financial interests."
- 11) A separate author contribution section is required following the Acknowledgments in all research manuscripts. All authors should be mentioned and designated by their first and middle initials and full surnames. We encourage use of the CRediT nomenclature (<https://casrai.org/credit/>).
- 12) ORCID IDs: ORCID IDs are unique identifiers allowing researchers to create a record of their various scholarly contributions in a single place. Please note that ORCID IDs are required for all authors. At resubmission of your final files, please be sure to provide your ORCID ID and those of all co-authors.
- 13) JCB requires authors to submit Source Data used to generate figures containing gels and Western blots with all revised manuscripts. This Source Data consists of fully uncropped and unprocessed images for each gel/blot displayed in the main and supplemental figures. Since your paper includes cropped gel and/or blot images, please be sure to provide one Source Data file for each figure that contains gels and/or blots along with your revised manuscript files. File names for Source Data figures should be alphanumeric without any spaces or special characters (i.e., SourceDataF#, where F# refers to the associated main figure number or SourceDataFS# for those associated with Supplementary figures). The lanes of the gels/blots should be labeled as they are in the associated figure, the place where cropping was applied should be marked (with a box), and molecular weight/size standards should be labeled wherever possible. Source Data files will be directly linked to specific figures in the published article.
- Source Data Figures should be provided as individual PDF files (one file per figure). Authors should endeavor to retain a minimum resolution of 300 dpi or pixels per inch. Please review our instructions for export from Photoshop, Illustrator, and PowerPoint here: <https://rupress.org/jcb/pages/submission-guidelines#revised>
- 14) Journal of Cell Biology now requires a data availability statement for all research article submissions. These statements will be published in the article directly above the Acknowledgments. The statement should address all data underlying the research presented in the manuscript. Please visit the JCB instructions for authors for guidelines and examples of statements at (<https://rupress.org/jcb/pages/editorial-policies#data-availability-statement>).

B. FINAL FILES:

Thank you for your attention to these final processing requirements. Please revise and format the manuscript and upload materials within 7 days. If you need an extension for whatever reason, please let us know and we can work with you to determine a suitable revision period.

Thank you for this interesting contribution, we look forward to publishing your paper in Journal of Cell Biology.

Sincerely,

Jodi Nunnari, PhD
Editor-in-Chief
Journal of Cell Biology

Dan Simon, PhD
Scientific Editor
Journal of Cell Biology

Reviewer #1 (Comments to the Authors (Required)):

I am glad to read the revised manuscript and the response letter, which have mostly cleared my concerns. I am not fully satisfied with the argument regarding the OsTIR1+/- issue commented on by the authors. However, I understand that the effect of GAL-OsTIR1 is weak, and the conclusions presented in this manuscript will not change.

Additionally, the companion paper by the Schuldiner group used yeast expressing OsTIR1 in all experiments and compared the cells treated with and without 5-Ph-IAA. In contrast, the paper by the Khmelinskii group described the effect of the N- and C-terminal tagging (Fig. 1). The two papers are complementary.

I feel that this manuscript is ready to be published in JCB.

Reviewer #2 (Comments to the Authors (Required)):

The revision has fully addressed all my concerns. I support the publication of this paper.

Reviewer #3 (Comments to the Authors (Required)):

The authors have done a nice job of addressing the reviewers' comments. The manuscript is now acceptable for publication.